# Descending motor circuitry required for NT-3 mediated locomotor recovery after spinal cord injury in mice

Qi Han[1,2], Josue D. Ordaz[1,2], Nai-Kui Liu[1,2], Zoe Richardson[1,2], Wei Wu [1,2], Yongzhi Xia[1,2], Wenrui Qu[1,2], Ying Wang[1,2], Heqiao Dai[1,2], Yi Ping Zhang[3], Christopher B. Shields[3,4], George M. Smith[5] & Xiao-Ming Xu [1,2]*

Locomotor function, mediated by lumbar neural circuitry, is modulated by descending spinal pathways. Spinal cord injury (SCI) interrupts descending projections and denervates lumbar motor neurons (MNs). We previously reported that retrogradely transported neurotrophin-3 (NT-3) to lumbar MNs attenuated SCI-induced lumbar MN dendritic atrophy and enabled functional recovery after a rostral thoracic contusion. Here we functionally dissected the role of descending neural pathways in response to NT-3-mediated recovery after a T9 contusive SCI in mice. We find that residual projections to lumbar MNs are required to produce leg movements after SCI. Next, we show that the spared descending propriospinal pathway, rather than other pathways (including the corticospinal, rubrospinal, serotonergic, and dopaminergic pathways), accounts for NT-3-enhanced recovery. Lastly, we show that NT-3 induced propriospino-MN circuit reorganization after the T9 contusion via promotion of dendritic regrowth rather than prevention of dendritic atrophy.

[1] Spinal Cord and Brain Injury Research Group, Stark Neurosciences Research Institute, Indiana University School of Medicine, Indianapolis, IN 46202, USA. [2] Department of Neurological Surgery, Indiana University School of Medicine, Indianapolis, IN 46202, USA. [3] Norton Neuroscience Institute, Norton Healthcare, Louisville, KY 40202, USA. [4] Department of Neurological Surgery, University of Louisville, Louisville, KY 40292, USA. [5] Department of Neuroscience, Shriners Hospitals Pediatric Research Center, Center for Neural Rehabilitation and Repair, Lewis Katz School of Medicine, Temple University, Philadelphia, PA 19122, USA. *email: xu26@iupui.edu

Locomotion is a universal and robust motor function that allows humans and other animals to interact with their surroundings[1]. Although locomotion appears to be rhythmic and repetitive, it is a complex motor behavior that involves spinal circuits at caudal regions and descending pathways from rostral centers[2]. In the spinal cord, motoneurons (MNs) are the final stage of neural processing for the execution of locomotor behaviors[3,4]. In the supraspinal areas, pathways from the cortex and brainstem contribute to the planning and initiation of locomotion[1,5]. The recruitment of both spinal and supraspinal motor circuits collectively produce and adapt the rhythms and patterns of locomotion with behavioral needs in immediate surroundings[6].

Spinal cord injury (SCI) not only disrupts descending pathways at the injury site but also results in MN degeneration with dendritic withdrawal or atrophy below the injury, collectively leading to impaired locomotor functions[7,8]. Clinically, ~67% of SCI patients are subject to an incomplete injury[9]. Imaging[10], electrophysiological[11], and anatomical evaluations[12,13] reveal that SCIs, even in the most severe cases, usually spare some region of the spinal cord. These spared neural tissues contain residual fibers from mixed supraspinal and propriospinal pathways that maintain a physical connection with the lumbar motor circuits which coordinate locomotion[14,15]. Therefore, the residual connections play important roles in facilitating locomotor recovery after SCI. For example, Leonie and colleagues showed that neuromodulation of residual reticulospinal projections after severe spinal cord contusion relayed the cortical command and enabled functional locomotor recovery[14].

In addition to supraspinal pathways, the propriospinal system is often partially intact following contusive SCI. Reorganization of propriospinal relay connections that bypass the injury sites are able to mediate spontaneous locomotor recovery[15–17]. Besides residual projections, studies have shown that neurotherapy and rehabilitation can change the phenotype of MNs and increase MN plasticity, leading to remodeling of lumbar motor circuits that control locomotor behaviors after SCI[8,18,19]. Thus, residual projections and lumbar MNs are two potential targets for therapeutic interventions that could enable locomotor recovery after SCI.

Neurotrophin-3 (NT-3), a member of the neurotrophic family of proteins, has been shown to play both growth-stimulating and chemo-attractive roles in motor restoration by promoting axonal growth and synaptic plasticity in multiple spinal pathways[20,21]. We recently reported that retrogradely transported NT-3 to the lumbar MNs significantly attenuated SCI-induced lumbar MN dendritic atrophy and improved locomotor recovery in adult mice after a moderate T9 contusive SCI[7]. However, the mechanism underlying recovery remains elusive. Is the extent of NT-3-induced recovery related to the preservation of lumbar MN dendrites or the degree of spared and/or sprouted descending pathways, or both? Moreover, if residual projections are involved, which descending pathways survive a contusion injury, respond to NT-3 treatment, and contribute to the reorganization of the lumbar neural circuit and thus to the recovery of locomotor function? Addressing these questions has considerable therapeutic implications for developing care and treatment of SCI.

In this study, we first demonstrate that residual descending pathways projecting to the lumbar region of spinal cord are essential for hindlimb locomotion restoration. We then show that a moderate thoracic contusion at T9 abolishes the corticospinal tract (CST) and rubrospinal tract (RST) projections down to the spinal cord, but retains partial neural transmissions downstream, which can be reinforced by the NT-3 therapy through residual descending propriospinal neurons (dPNs) and their tract (dPST). With anatomical and physiological analysis and pathway-selective silencing, we also reveal that the spared propriospinal pathway, rather than other descending pathways, is functionally associated with NT-3-mediated locomotor recovery after SCI. Quantitative analysis of time-dependent dendritic ramifications of lumbar MNs suggests that NT-3 mediates MN recovery via promoting dendritic regrowth (a neuroplasticity mechanism) rather than by preventing dendritic atrophy (a neuroprotective mechanism).

## Results

**Descending pathways are required in NT-3 mediated recovery.** Our previous results demonstrated that retrograde transport of adeno-associated virus-NT-3 (AAV-NT-3) to lumbar MNs partially restored lumbar motor circuitry, including both lumbar MNs and surrounding residual descending fibers, leading to enhanced locomotor recovery after T9 contusion[7]. However, it remains unclear whether lumbar MNs and spared descending pathways work alone or together to induce NT-3-mediated recovery. To investigate the involvement of descending pathways, we first conducted a complete spinal transection at T9 spinal cord and injected either adeno-associated virus–green fluorescent protein (AAV-GFP) or a mixture of AAV-GFP and AAV-NT-3 (1:1 ratio) through the pre-demyelinated sciatic nerves for retrograde transport of GFP or NT-3 to lumbar MNs (Fig. 1a). Six weeks post-injury (wpi), we found that the T9 transection created a complete lesion gap (GFAP-negative) in the spinal cord that severed all descending projections, including the biotinylated dextran amine (BDA)-labeled dPST, to the spinal cord below T9 segment (Fig. 1a). Consequently, lumbar MNs caudal to the T9 transection underwent degeneration with profound dendritic atrophy (Fig. 1b). Notably, the retrograde transport of AAV-NT-3 to the lumbar MNs significantly attenuated MN degeneration by increasing their dendritic complexity (Fig. 1b, c). However, such lumbar MN recovery after complete transection in NT-3-treated mice failed to show significant functional improvement in Basso-Mouse-Score (BMS) test, as compared to non-NT-3-treated mice (Fig. 1d). This result indicates that the recovery of lumbar MNs alone is insufficient to restore meaningful locomotion, thus the recruitment of spared descending motor pathways down to the lumbar MN pool remains critical for functional recovery after SCI.

**NT-3 treatment enhances supraspinal transmission after SCI.** Cognizant of the potential role of descending pathways in recovery, we next investigated which of the spared supraspinal pathways contributed to NT-3-induced locomotor recovery after a thoracic contusive SCI and the potential mechanism[7]. We first investigated the possible connectivity of cortico-MN and rubro-MN circuits after SCI. Electrical stimulation in the motor cortex or red nucleus elicited reproducible waveforms of electromyography (EMG) responses in sham mice (Fig. 2a, c). After the T9 contusion, small yet detectable reflex EMG signals were still evoked in AAV-GFP-treated mice after either motor cortex or red nucleus stimulation (Fig. 2a, c). Notably, AAV-NT-3 treatment elevated the EMG activity of hindlimb muscles, as reflected by increased peak-to-peak amplitudes and decreased signal latencies relative to those in AAV-GFP treated controls (Fig. 2a, c, e, f). This indicates that NT-3 treatment enhanced the transmission of supraspinal signals across the contusion to activate the lumbar circuit. When we re-lesioned the CST by bilateral pyramidotomy in the medulla, or re-lesion the RST by bilateral C2 lateral hemisection at levels rostral to the T9 contusion, no EMG responses were elicited in all groups at 1 day after the re-lesion even with gradually increased electrical stimulation (Fig. 2b, d). This suggests that the transmission examined in sham and contusive mice was mainly due to spared supraspinal transmission to

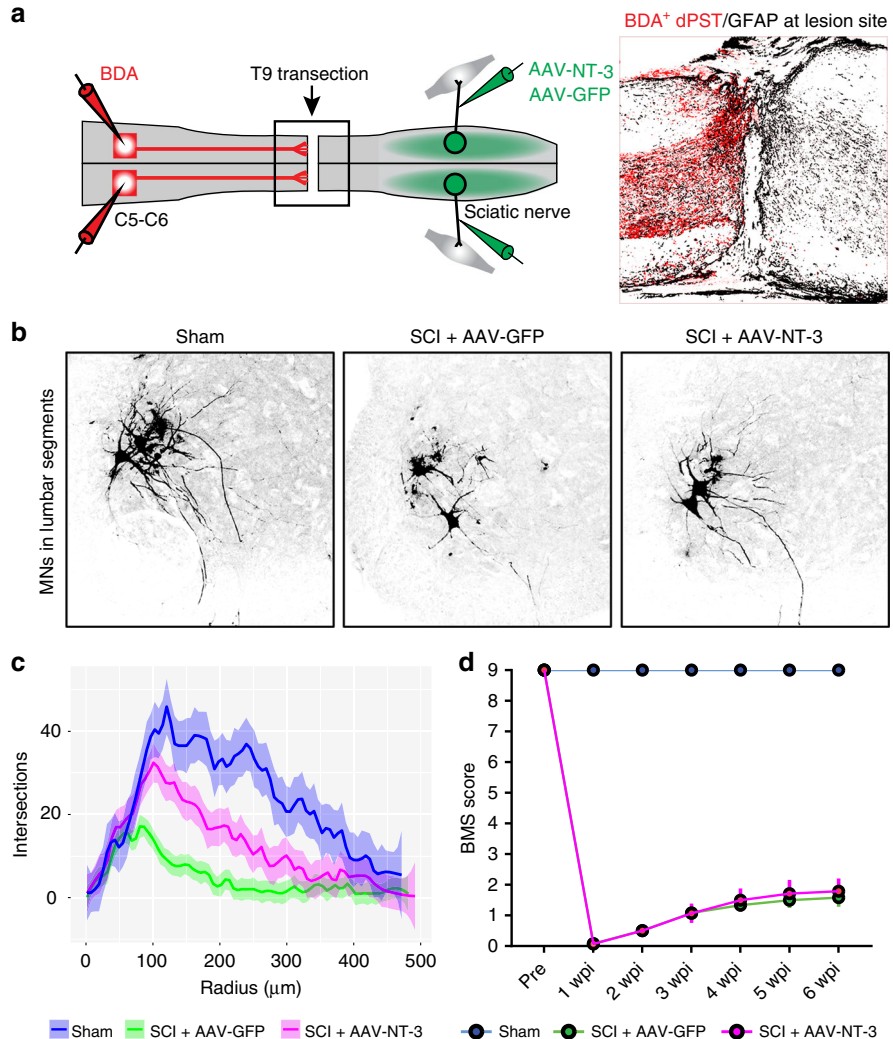

**Fig. 1 NT-3 failed to improve functional recovery after T9 transection. a** Diagram illustrates the experimental design. A complete spinal cord transection was made at the T9 vertebral level to transect all descending pathways. AAV-NT-3 or AAV-GFP (control) was injected bilaterally into sciatic nerves to retrogradely transport the NT-3 to lumbar MNs. BDA was bilaterally injected into cervical segments of C5–C6 to anterogradely label the cervical descending dPST at 6 wpi. A representative image of the boxed area (right) shows the lesion site, recognized by GFAP staining, in which BDA-labeled (BDA$^+$) dPST axons were completely stopped at the rostral lesion border. Scale bar = 200 μm. Compass: R, rostral; C, caudal; D, dorsal; V, ventral. **b** Representative images show lumbar MN dendritic complexity from sham and T9 transected mice treated with either AAV-GFP or AAV-NT-3. Scale bar = 200 μm. **c** Curve plot represents the sholl analysis of lumbar MN dendritic distribution in each experimental group. Thick lines represent the average distribution of $n = 3$–4 animals per group; shaded area, 95% confidence intervals. **d** Line plot indicates the BMS score in three experimental groups, before and after the T9 transection, at different time points. Note that mice receiving AAV-NT-3 treatment after the T9 transection show increased MN dendritic arbors, but no locomotor improvement as compared to mice receiving AAV-GFP. Data are presented as mean ± SEM; $n = 5$–7 biologically independent animals per group. Two-way ANOVA followed by Tukey's multiple comparison tests. AAV adeno-associated virus, NT-3 neurotrophin-3, GFP green fluorescent protein, BDA biotinylated dextran amines, dPST descending propriospinal tract, GFAP glial fibrillary acidic protein, MNs motoneurons. Source data are provided as a Source Data file.

the lumbar motor circuit, rather than autonomous activities of the lumbar MN circuit.

**Contusion at T9 disrupts CST and RST projections below the lesion site.** The electrophysiological evaluation showed that descending signals from both CST and RST could be used to evaluate lumbar circuits during NT-3 treatment. These results prompted us to assess the possible integrity of CST and RST projections after T9 contusion. We began by injecting an anterograde tracer, BDA, into the sensorimotor cortex at 6 wpi (Supplementary Fig. 1a). In sham mice, BDA-labeled CST axons projected to the lumbar enlargement bilaterally and their

terminals distributed extensively in the gray matter on both sides (Supplementary Fig. 1b, c). Some terminals were in close contact with cholera toxin subunit B (CTB)-labeled lumbar MNs and co-localized with synaptophysin puncta, indicating direct cortico-MN connections in adult mice (Supplementary Fig. 2a, b). However, few, if any, BDA-labeled CST fibers were observed caudal to the lesion in either AAV-GFP- or AAV-NT-3-treated mice (Supplementary Fig. 2a, b). The lack of CST projections caudal to the injury is clearly illustrated in heat maps generated from sections between L2–L4 segments (Supplementary Fig. 1c). Serial sections showed that BDA-labeled CST axons projected extensively above the injury, diminished at the site of injury, and disappeared below the level of injury (Supplementary Fig. 3a).

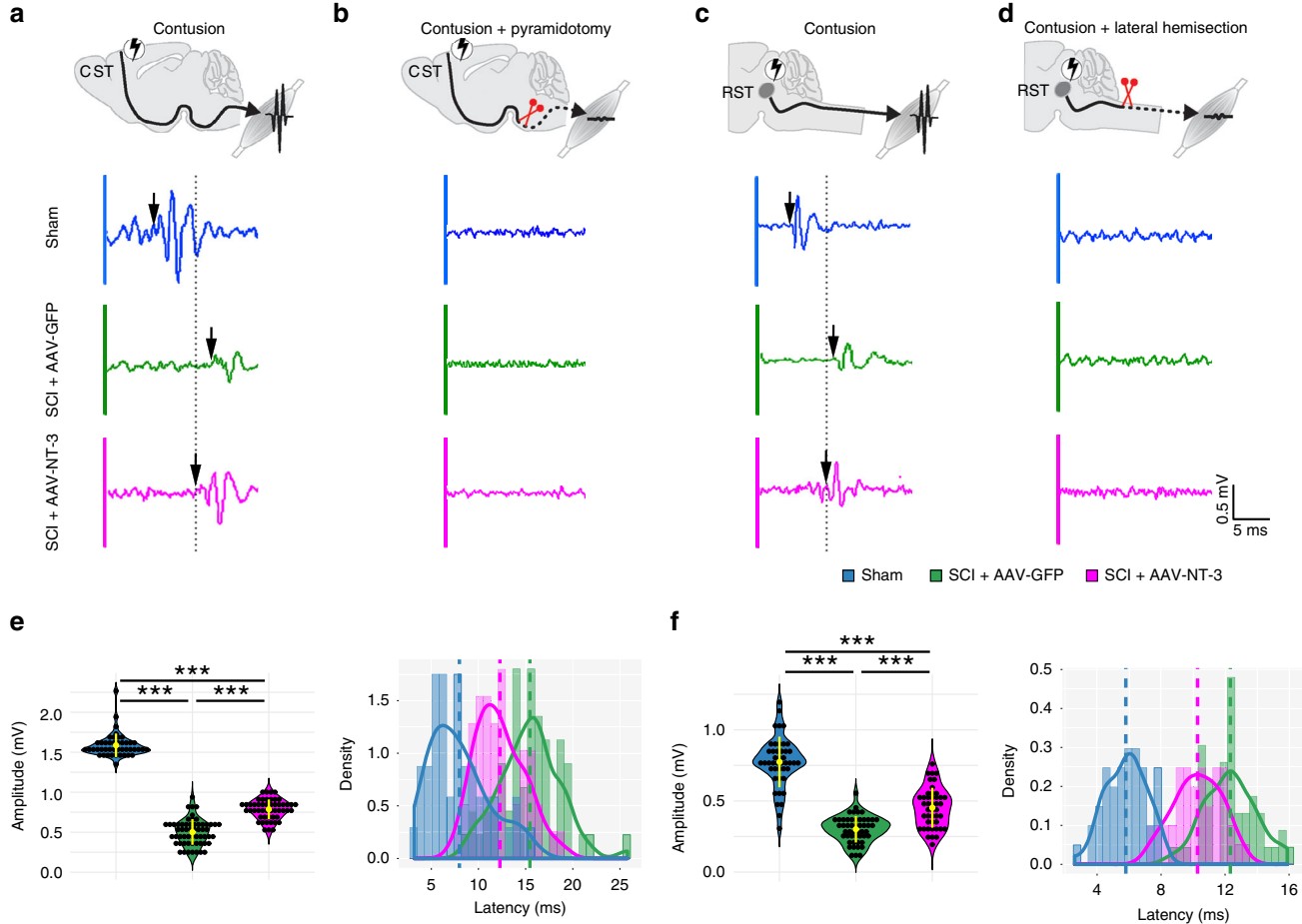

**Fig. 2 NT-3 enhanced cortical and rubral control of hindlimb EMG response. a** Schematic diagram shows EMG potentials recorded in contralateral gastrocnemius muscle in response to electrical stimulation in the motor cortex of mice at 6 weeks after a T9 contusion. Representative examples of EMG traces were evoked by motor-cortex stimulation in the sham, SCI + AAV-GFP, and SCI + AAV-NT-3 treatment groups after SCI. The dotted line indicates the onset of EMG response in an AAV-NT-3-treated mouse; the arrows indicate the EMG latency. **b** Schematic diagram illustrates EMG activity in each group at 1 day after bilateral pyramidotomy. **c** Schematic diagram shows EMG potentials recorded in the contralateral gastrocnemius muscle in response to electrical stimulation in the red nucleus of mice at 6 weeks after the T9 contusion. Representative examples of EMG traces were plotted for each animal group. **d** Schematic diagram illustrates the EMG activities in each animal group at 1 day following bilateral lateral hemisection to transect the RST at the C5 vertebral level. **e** Quantitative analysis of cortico-stimulation-induced EMG amplitudes and latencies of the sham (44 stimulation sites from 6 mice), SCI + AAV-GFP (57 stimulation sites from 8 mice), and SCI + AAV-NT-3 (50 stimulation sites from 7 mice) animal groups. **f** Quantitative analysis of rubro-stimulation-induced EMG amplitudes and latencies in the sham (44 stimulation sites from 5 mice), SCI + AAV-GFP (49 stimulation sites from 6 mice), and SCI + AAV-NT-3 (45 stimulation sites from 6 mice) animal groups. The black dots represent independent stimulation sites from each group. The yellow lines from violin plots indicate mean ± SD. The dashed lines in histograms indicate the mean line from each group. ***$P < 0.001$. One-way ANOVA followed by Tukey's multiple comparison test. EMG electromyography, CST corticospinal tract, RST rubrospinal tract. Source data are provided as a Source Data file.

Thus, a moderate contusive SCI at T9 completely removes CST projections caudal to the injury.

To test the continuity of RST axons after T9 contusion, we microinjected BDA stereotaxically into the magnocellular region of the red nucleus (RN) bilaterally at 6 wpi to label the RST axons (Supplementary Fig. 1d, e). Surprisingly, we found that the moderate contusion damaged nearly all of the RST projections at and below the level of injury, even though the BDA-labeled RST located in the dorsolateral funiculus of spinal cord rostral to the lesion. Few, if any, BDA-labeled RST terminals were present at levels caudal to the T9 contusion in either AAV-GFP or AAV-NT-3 treated mice (Supplementary Figs. 1f, 2c, d, and 3b). Together, we found no evidence for the presence of either CST or RST axons directly projecting into the spinal cord caudal to the injury. Thus, functional connections between these pathways and lumbar neural circuitry must be mediated through intermediate pathways, such as the dPST, after SCI.

**NT-3 treatment induces propriospino-MN reorganization.** The first candidate for such an intermediate pathway to relay the CST and/or RST connectivity is the dPST, which is distributed uniformly over a wide area throughout the spinal cord[22,23], and serves as an intermediate relay for descending motor commands to activate the lumbar circuit[15,16,24]. To examine this possibility, BDA was injected into the rostral cervical segments (C5–C6) in mice to label the dPST at 6 wpi in each group (sham, SCI + AAV-GFP, and SCI + AAV-NT-3) (Fig. 3a). We found that unlike the CST and RST, a contusive SCI caused damage to the dPST but spared its projects to the lumbar spinal cord (Fig. 3b, c, Supplementary Fig. 4a–d). Notably, retrograde transport of AAV-NT-3 to lumbar MNs did not alter dPST fiber density in the thoracic level (Supplementary Fig. 4c, d), but significantly increased the terminal fiber density in lumbar spinal cord, especially in the ventral horn, where lumbar MNs reside, as compared to non-NT-3 treatment (Fig. 3b, c, d). Accordingly, a higher density of dPST

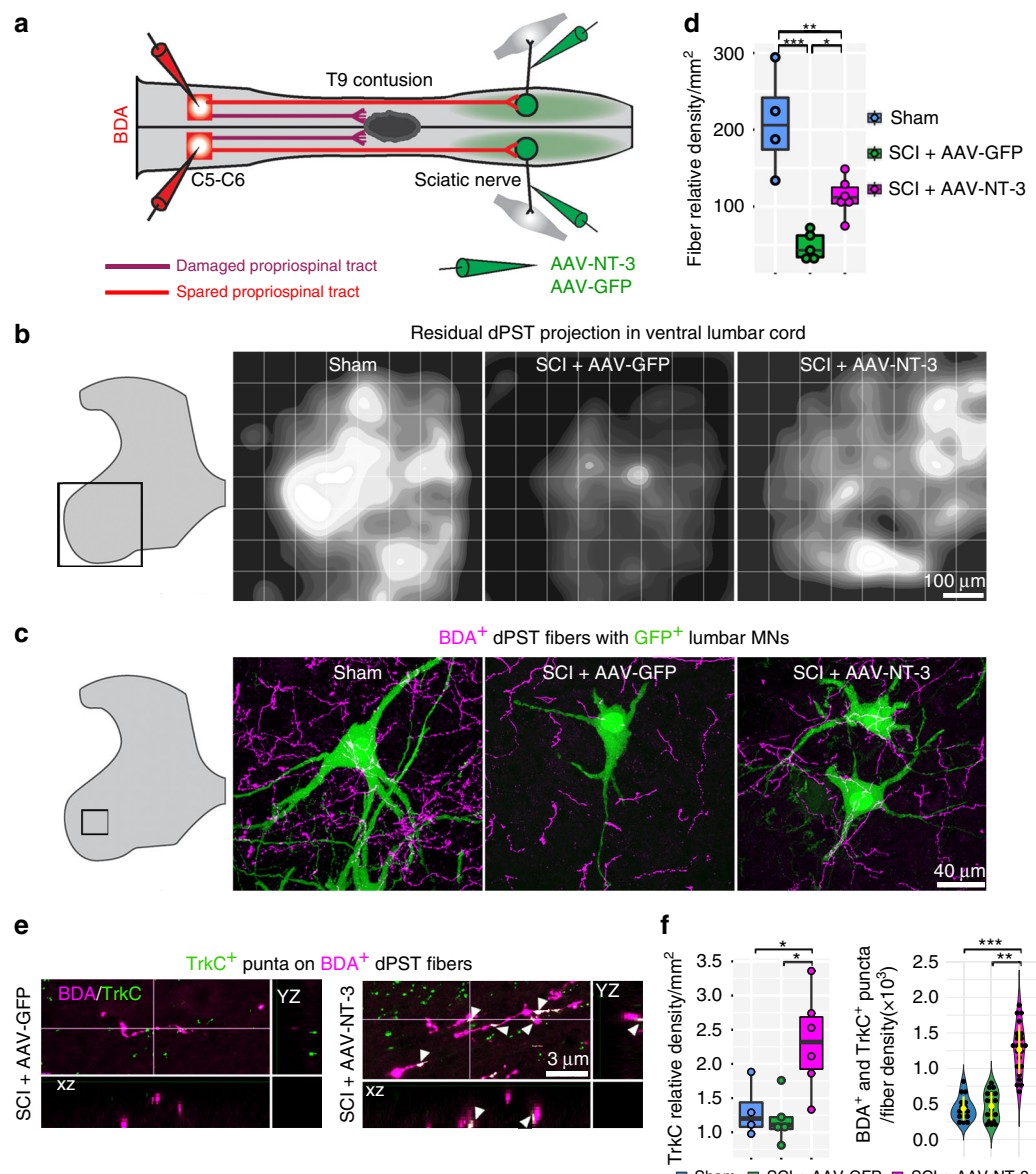

**Fig. 3 Residual dPST responded to NT-3 by increasing its terminal sprouting and TrkC expression. a** Diagram illustrates BDA injections into cervical segments (C5–C6) to label the cervical dPST axons projecting to the lumbar spinal cord following the T9 contusion. **b** Representative black-to-white intensity scales demonstrating residual dPST axonal distribution in the ventral horn of the lumbar spinal cord (boxed area) after the T9 contusion in each experimental group. **c** Higher magnification of boxed area shows close apposition of BDA+ dPST axons with lumbar MNs (GFP+), which were more frequently seen in the SCI + AAV-NT-3 group than the SCI + AAV-GFP group. Scale bar = 40 μm. **d** Bar plot indicates the relative integrated density/mm$^2$ of BDA+ dPST fibers in the lumbar ventral horn between groups; $n = 4$–6 mice per group (3–5 sections per animal). **e** Confocal z-stacks demonstrate that TrkC was expressed in the lumbar spinal cord, with some TrkC co-localization with BDA+ dPST terminals in the AAV-NT-3-treated group (arrowheads). Scale bar = 3 μm. **f** Bar plot shows TrkC relative integrated density/mm$^2$ and individual values for each animal (3–5 sections per animal). Violin plot represents the ratio of double-labeled puncta (TrkC and BDA) to BDA relative integrated density in the corresponding sections. The yellow lines from the violin plots indicate mean ± SD. Data are presented as box plots with center lines indicating medians, boxes representing 25th to 75th percentiles, also known as the interquartile range (IQR), and whiskers representing data points within 1.5 times the IQR; $n = 4$–6 biologically independent animals per group. *$P < 0.05$; **$P < 0.01$; ***$P < 0.001$. One-way ANOVA followed by Tukey's post-hoc test. dPST descending propriospinal tract, TrkC tropomyosin receptor kinase C. Source data are provided as a Source Data file.

terminals contacting lumbar MNs appeared in SCI + AAV-NT-3 than in SCI + AAV-GFP groups (Fig. 3c, d, Supplementary Fig. 4e, f). Importantly, AAV-NT-3 treatment significantly enhanced TrkC (a highly selective receptor for NT-3) expression in the gray matter of lumbar spinal cord and accordingly elevated TrkC-BDA colocalization along with BDA-labeled dPST axons, as compared to AAV-GFP-treated controls (Fig. 3e, f).

To confirm the NT-3-mediated propriospino-MN circuit reorganization, a retrograde multi-transsynaptic pseudorabies virus (PRV) carrying the enhanced GFP (EGFP) was introduced to retrogradely label the lumbar MN-related interneurons above, at, and below the lesion (Supplementary Fig. 5a). PRV labels different-order neurons in a time-dependent manner[25]. Therefore, we first detected PRV-labeled lumbar MNs at 24 h post-injection (Supplementary Fig. 5b). At 72 h, viral propagation is sufficient to identify the distribution of dPNs as we observed PRV-labeled interneurons located in the cervical level (Supplementary Fig. 5c). In sham mice with PRV injection, GFP-positive

interneurons were distributed widely throughout the spinal cord, mostly in the dorsal horn and intermediate zone. In contrast, contusion injury dramatically reduced the population of retrogradely labeled interneurons above the lesion, where they were found limited to intermediate laminae[26]. However, there was a significant increase in PRV-labeled dPNs at the T7 segmental level in SCI mice with AAV-NT-3 treatment, as compared to SCI mice with AAV-GFP treatment (Supplementary Fig. 5c, d). These results suggest that NT-3 treatment reinforced residual propriospino-MN circuit reorganization by increasing their synaptic connections.

**The dPNs relay descending signals bypassing the lesions**. To identify whether the dPNs above the lesion could relay descending commands down to the lumbar circuit, we need to first confirm whether these pathways make connections with the dPNs at levels above the injury. In an initial attempt, we determine the projection of three descending pathways originating from the motor cortex (forms the CST), red nucleus (forms the RST), and cervical enlargement (forms the cervical dPST) on dPNs above the level of injury. We injected the anterograde tracer, BDA, into the motor cortex (Supplementary Fig. 6a), red nucleus (Supplementary Fig. 6d), and cervical spinal cord (C5–C6, Supplementary Fig. 6g), respectively. At the same time, we retrogradely traced lumbar MN-related dPNs at T7 (rostral to the T9 contusion) by injection of PRV into the gastrocnemius muscle (Supplementary Fig. 6a, d, g). In the intact spinal cord, many BDA-labeled axons from all of the three descending pathways terminated on PRV-labeled dPNs at T7 (Supplementary Fig. 6b, e, h), indicating the establishment of cortico-, rubro-, and propriospino-MN circuits within the spinal cord rostral to a T9 contusion, which are important to propagate descending signals down to lumbar MNs. Although a T9 contusion injury reduced the number of GFP-positive neurons rostral to the injury, the residual PRV-labeled dPNs still made close contacts with BDA-labeled CST, RST, and cervical dPST, which were colocalized with synaptophysin, a presynaptic marker (Supplementary Fig. 6c, f, i). Together with the electrophysiology data (Fig. 2), these findings suggest that residual dPST that passes around the injury serves as an anatomical bridge to relay supraspinal signals to lumbar MN circuit after a T9 contusion, and these axons responded to the NT-3 treatment.

**Silencing dPNs impairs NT-3-induced locomotor recovery**. To further determine whether descending propriospino-lumbar MN circuit reorganization was responsible for NT-3-mediated enhancement of recovery of function, we reversibly silenced MN-related dPNs by selective blockades of the dPN transmission via a dual-viral infection technique[23] (Supplementary Fig. 7a), a method originally developed by Isa and colleagues[27]. Here, we chose to functionally silence the neurotransmission of thoracic dPNs rather than cervical ones due to the lack of a significant number of PRV-labeled dPNs in cervical levels in contusive mice (Supplementary Fig. 5c). In this study, the first vector, HiRet-TRE-EGFP.eTeNT carrying the EGFP and enhanced tetanus neurotoxin light chain (eTeNT) downstream of the tetracycline-responsive element (TRE), was bilaterally injected into the L2–L4 cord level where propriospinal axon terminals innervate the lumbar MN region (Fig. 4a, b). The second vector, Tet-On/AAV, carrying the Tet-on sequence, a variant of reverse tetracycline transactivator (rtTAV16), under the control of the cytomegalovirus (CMV) promoter, was bilaterally injected into the intermediate zone of T5–T7 spinal cord to infect thoracic dPNs (Fig. 4a, b). Three weeks after the injections, doxycycline (Dox) was given to mice to induce the expression of EGFP-tagged

eTeNT in dPNs which were transduced by both vectors. eTeNT would selectively silence synaptic transmission from dPNs without perturbation of other descending neurotransmissions to lumbar MNs (Fig. 4a, b).

The double-infected dPNs, visualized by EGFP expression, were found mainly in the lateral portion of the intermediate zone (laminae V–VII) (Fig. 4b), which were similar to the distribution of PRV-labeled dPNs observed previously. Their longitudinal distribution extending from T4 to T8 segments was slightly wider than where Tet-On/AAV was injected (T5–T7) and probably resulted from the diffusive spread of the injected vector solution. Contusion injuries did not affect dPN distributions in the spinal cord, but markedly reduced the number of EGFP-positive dPNs by 60% when compared with that in sham mice (Fig. 4b). No significant difference was found in the number of dPNs between contusive mice with either AAV-GFP or AAV-NT-3 treatment (Fig. 4b). To determine the functional (i.e. locomotion) consequences following the dPNs blockage, we performed time-course behavioral assessments (BMS, grid walking, and rotarod tests) on each experimental group before, during, and after oral Dox administration. After beginning Dox administration, sham mice expressing eTeNT had profound deficits of skilled locomotion, as revealed by increased error rate in grid walking and reduced ability to stay on the rod in the rotarod test (Fig. 4c), indicating dPN-mediated transmission was associated with hindlimb locomotion in the intact spinal cord. No significant changes were observed in the BMS score during the silencing of dPNs (Supplementary Fig. 7b). This may be because blockage of synaptic transmission through partial thoracic dPNs is not severe enough to cause behavior deficits in the BMS test, or because other descending pathways would have compensated for the loss of control by silenced dPNs. In SCI + AAV-GFP control mice, Dox administration caused detectable but not significant locomotor deficits in grid walking and rotarod tasks (Fig. 4c). In contrast, silencing dPNs in SCI + AAV-NT-3 mice significantly impaired the skilled locomotor movements in both grid walking and rotarod tests which had previously shown recovery with NT-3 treatment (Fig. 4c). This result was substantiated when we analyzed the locomotor performance of individual animals before, during, and after Dox administration (Fig. 4c). Furthermore, we found a higher positive correlation between the silencing-induced locomotion alterations and the number of double infected dPNs in the SCI + AAV-NT-3 group, compared to the poor correlation in the sham or SCI + AAV-GFP group (Fig. 4d).

To further confirm whether the descending propriospinal reorganization is responsible for NT-3-mediated enhanced functional recovery, we examined mice that had received the NT-3 treatment after two, spatially separated hemisections at T7 and T12, on opposite sides of the spinal cord, to test the effect of transecting all descending axons while leaving those contiguous propriospinal circuits (Fig. 5a, b). We unilaterally injected BDA into the right side of the spinal cord at T9 at 6 weeks after the staggered lesions and found a significant increase in fiber density of BDA-labeled dPST sprouting from right to left side of the lumbar segments in AAV-NT-3-treated mice, relative to AAV-GFP-treated control mice (Fig. 5c). In accordance, more dPST fibers were found in close apposition with CTB-labeled lumbar MNs at the left lumbar segments in AAV-NT-3-treated mice as compared to the AAV-GFP controls (Fig. 5d). Consistent with the anatomical differences, the lesioned mice with AAV-GFP treatment had limited left hindlimb movement even at 6 wpi, and the left gastrocnemius muscle was quiescent after transcranial magnetic motor evoked potential (tcMMEP) stimulation (Fig. 5e). However, the lesioned mice that received NT-3 treatment had some extent of recovery in stepping ability, and the gastrocnemius

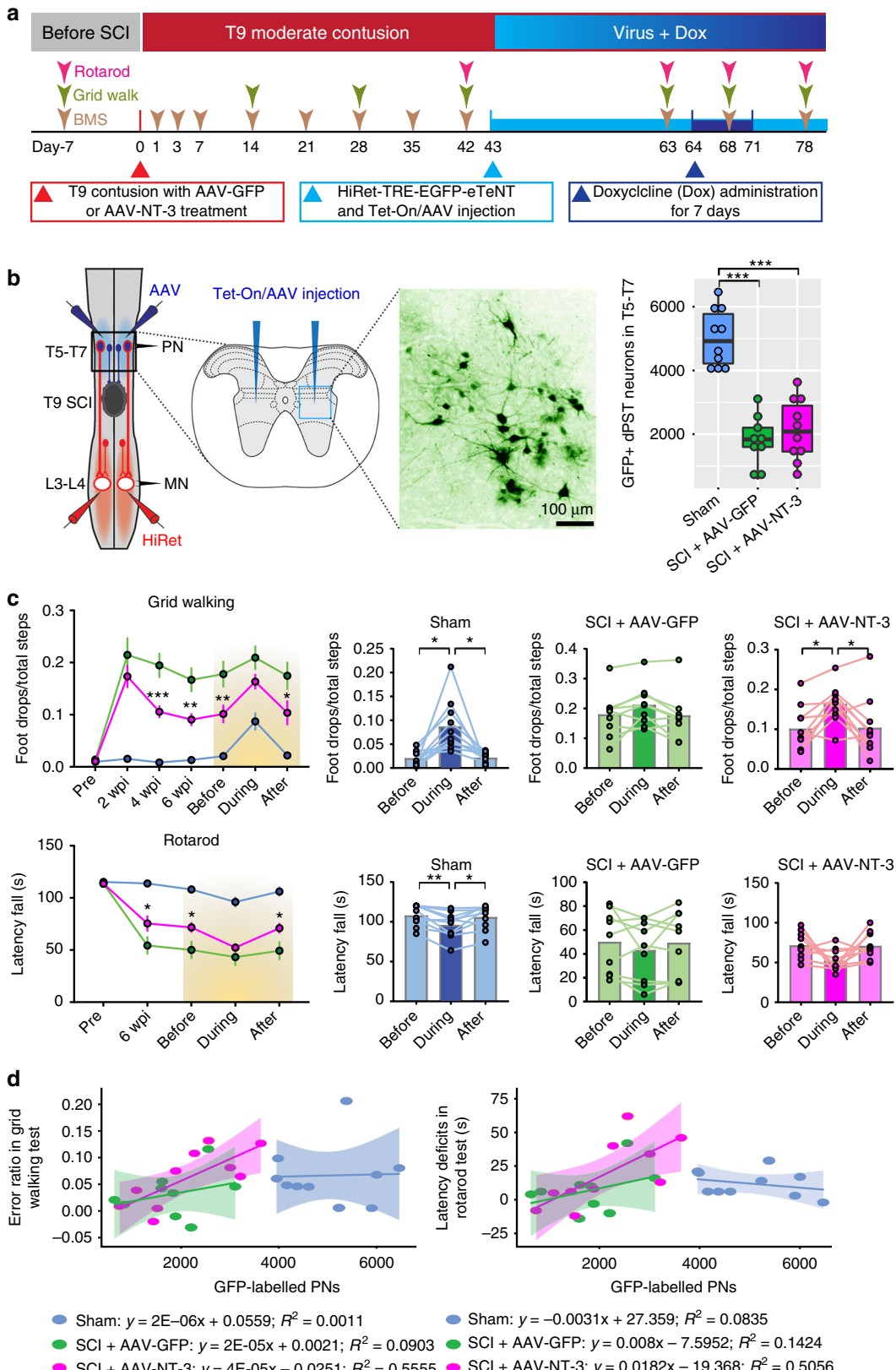

muscle in the left hindlimb showed consistent and detectable tcMMEP responses, despite abnormal latency and amplitude, as compared to sham intact mice (Fig. 5f). Collectively, these findings suggest that the descending propriospino-lumbar MN circuit reorganization is functionally required for NT-3-induced functional recovery.

**Silencing monoaminergic pathways produces no locomotor deficits**. In addition to descending propriospinal pathways, AAV-NT-3 transfection also enhanced serotonin (5-HT, i.e. serotoninergic) and tyrosine hydroxylase (TH, i.e. dopaminergic) fiber density in the lumbar spinal cord (Supplementary Fig. 8), which corroborated our previous report using a similar

**Fig. 4 Thoracic dPNs functionally contributed to NT-3-mediated locomotor recovery. a** A schematic drawing shows the experimental design. The contusive mice with either AAV-GFP or AAV-NT-3 treatment received two viral injections, i.e. HiRet-TRE-EGFP-eTeNT into the L3–L4 segments and Tet-On/AAV into the T5–T7 segments at 6 wpi. Three weeks later, Dox was administered to induce the expression of eTeNT. Locomotor function was evaluated using BMS, grid walking, and rotarod assessments before, during and after Dox administration. **b** A diagram shows the strategy to silence the thoracic (T5–T7) dPNs with dual virus injections. Insets of the boxed area indicate the transverse view of Tet-On/AAV injections and a representative image of GFP+ dPNs infected by dual viruses. Scale bar = 100 μm. Bar plot reveals the total number of dual-virus infected, GFP+ dPNs in T5–T7 spinal segments among three groups with individual values for each animal. Data are presented as box plots with center lines indicating medians, boxes representing 25th to 75th percentiles, and whiskers representing data points within 1.5 times the IQR. n = 9–10 animals per group. ***$P < 0.001$; one-way ANOVA followed by Tukey's post-hoc test. **c** Line plots show changes of hindlimb locomotor function with grid walking and rotarod testing before, during and after Dox administration. Data are presented as mean ± SEM. n = 9–10 biologically independent animals per group. *$P < 0.05$, **$P < 0.01$, ***$P < 0.001$ (SCI + AAV-GFP vs SCI + AAV-NT-3); two-way ANOVA followed by Tukey's multiple comparisons test. Bar plots detail alterations in locomotor behaviors, with individual animals from each experimental group, in response to Dox treatment (n = 9–10 animals per group). *$P < 0.05$; **$P < 0.01$; one-way ANOVA followed by Tukey's post-hoc test. **d** Scatter plots indicating the correlations of the number of double-infected GFP+ dPNs with Dox-induced locomotor deficits in grid walking and rotarod assessments. The middle lines in scatter plots indicate the regression lines and the shades represent 95% confidence intervals. SCI spinal cord injury, BMS Basso Mouse Scale, Dox doxycycline, wpi weeks post-injury. Source data are provided as a Source Data file.

approach[7]. Studies have shown that both descending serotoninergic and dopaminergic pathways are implicated in the initiation of locomotion[28–31]. Nevertheless, a difficult question remains: are these pathways functionally related to NT-3-induced locomotor recovery? To answer this, we used a similar double-viral blockage technique as we did for silencing dPNs. To selectively target the serotonin-MN circuit, we performed stereotaxic microinjections of HiRet-TRE-EGFP.eTeNT into the lumbar MN area (L2–L4) innervated by serotonergic axon terminals, and Tet-On/AAV into the caudal raphe nuclei where the 5-HT neurons are located (Supplementary Fig. 9a). More than 1000 double-transfected GFP-positive cells were found in the caudal brainstem in sham mice while less than 300 GFP-positive neurons were detected in the T9 contusive mice. No significant difference was noted in the number of GFP-positive neurons between SCI + AAV-GFP and SCI + AAV-NT-3 groups (Supplementary Fig. 9a, b). We used 5-HT immunohistochemistry to confirm the expression of serotonin in these cells. Any double-labeled (i.e. GFP and 5-HT) neurons would represent a serotoninergic neuron whose axon has projected to the lumbar cord. We found ~200 GFP-positive cells colocalized with 5-HT immunoreactive neurons from the raphe nuclei in sham mice but only ~50 GFP-labeled neurons expressed 5-HT in the raphe nuclei after the T9 contusion regardless of the treatment (Supplementary Fig. 9a, b). With 4 days of Dox administration, the mice in the sham group experienced a transient and slight motor deficit in grid walking test, but no detectable change in the rotarod test. Despite this deficit, contusive mice with either AAV-NT-3 or AAV-GFP treatment showed no problematic change in locomotion after silencing 5-HT projections (Supplementary Fig. 9b).

To selectively target the dopamine-MN circuit, the HiRet-TRE-EGFP.eTeNT was injected into lumbar MN region, and the Tet-On/AAV was stereotaxically injected into the A11 region within the posterior diencephalon, which has been known to give rise to descending dopaminergic axons and contribute to motor control[25,29,32] (Supplementary Fig. 9c). The double viral infection was confirmed by counting the GFP-labeled cells in the A11 region. We found that ~500 and ~90 GFP-positive cells were detected in the sham and T9 contusion mice, respectively. No difference in the number of GFP-positive neurons was found between the SCI + AAV-GFP and SCI + AAV-NT-3 groups. We did TH immunohistochemistry to identify classes of GFP-labeled cells within the A11 and found ~70 and ~15 GFP-positive cells were also positive for TH immunoreactive in sham and contusive mice, respectively. No difference in the number of GFP and TH double-infected neurons was found between SCI + AAV-GFP and SCI + AAV-NT-3 groups (Supplementary Fig. 9c, d). Similar

to the effect of blockage of serotoninergic transmission, A11 inactivation failed to induce the impairment of hindlimb locomotion between AAV-GFP and AAV-NT-3 treatments (Supplementary Fig. 9d). Even in sham mice, no change was found in locomotor behavior following Dox administration. Collectively, these results suggest that the spared serotoninergic and dopaminergic pathways play a minor role in NT-3-mediated enhancement of locomotor recovery after contusion.

**NT-3 reverses MN degeneration via dendritic regrowth rather than stabilization.** To better understand how AAV-NT-3 treatment mediates remodeling of propriospino-MN circuit following SCI, we first examined NT-3 expression by ELISA assay in the lumbar cord (L2–L4) at different time points following sciatic nerve AAV-NT-3 injections after the T9 contusion. Time-dependent NT-3 expression was significantly increased at 3 wpi, plateaued at 4 wpi, and maintained high levels of expression up to 6 wpi, the longest time point studied (Fig. 6a). We then chose 2 wpi and 4 wpi as two critical time points to examine NT-3-mediated remodeling of the propriospino-MN connections. The lumbar MNs were retrogradely labeled with CTB, while the dPST axons were anterogradely labeled with BDA (Fig. 6b). We found that NT-3 treatment had little effect on SCI-induced MN dendritic atrophy and synaptic stripping (of dPST axons) at 2 wpi, but promoted MN dendritic regrowth and new synaptic formation (from dPST axons) at 4 wpi (Fig. 6b, c, d). This suggests that the NT-3 effect on MN neurocircuit changes is a result of stimulating dendritic regrowth and synaptic reformation rather than stabilizing the existing dendritic morphology and synapses. However, such an explanation could still be challenged by the fact that NT-3 expression was relatively low at 2 wpi as compared to 4 wpi. To address this, we treated the mice with AAV-NT-3 at 2 weeks prior to the T9 contusion and then assessed MN dendritic changes at 2 wpi and 4 wpi. We found that, at 2 wpi, MN dendritic atrophy was still present and NT-3 pre-treatment had little effect on preventing it. At 4 wpi, however, MN dendritic ramifications appeared regrowing to a similar extent as with the NT-3 post-treatment (Fig. 6e). Thus, a similar pattern of MN dendritic atrophy occurred at 2 wpi followed by dendritic restoration at 4 wpi, irrespective of when NT-3 was given (pre- or post-injury), indicating that the NT-3 effect is mediated via promoting regrowth of atrophied dendrites rather than preventing existing dendrites from injury-induced dendritic atrophy. Likewise, NT-3 treatment also promoted the synaptogenesis of sprouted dPST axons and dendrites with lumbar MNs.

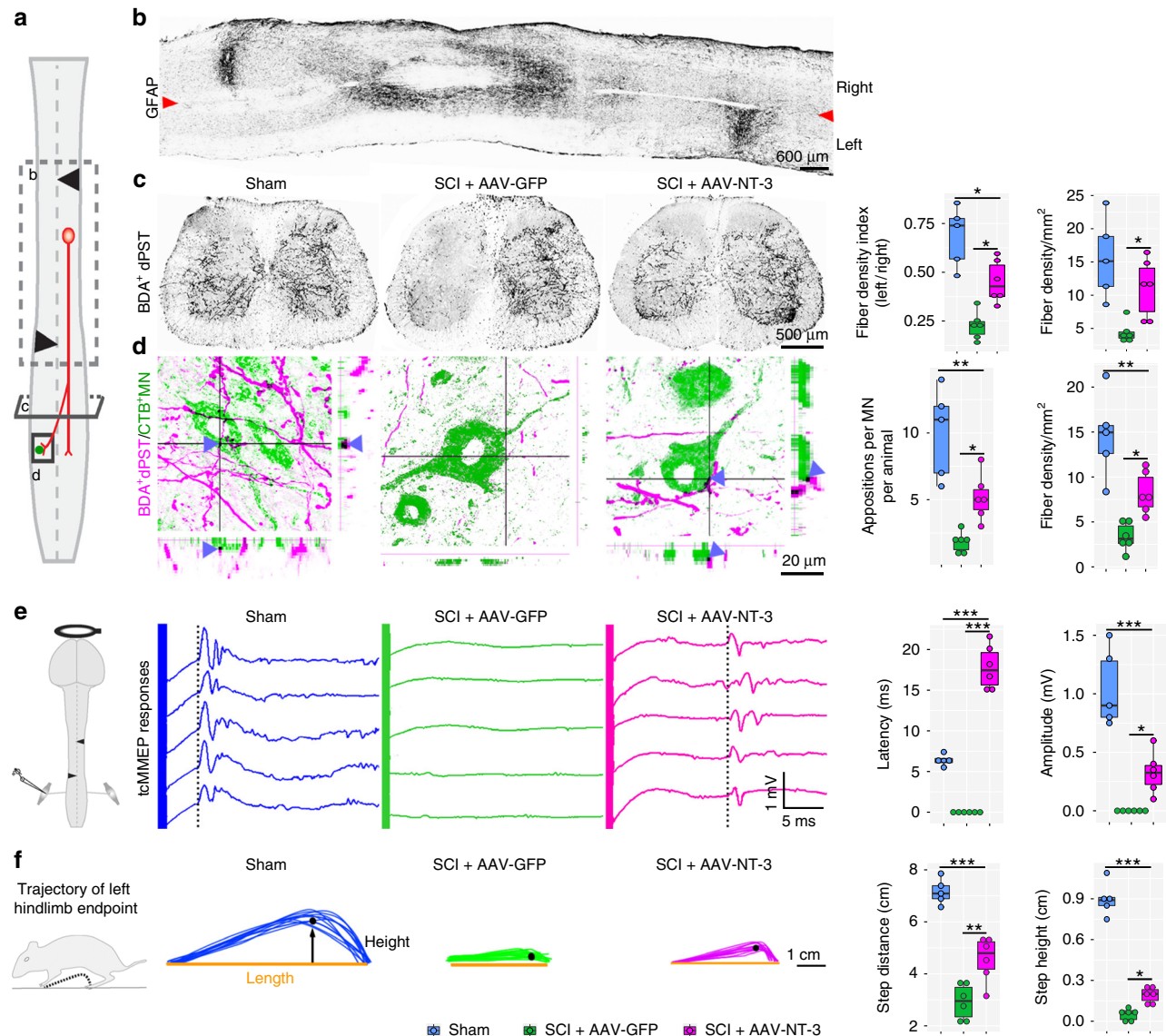

**Fig. 5 NT-3 promoted propriospino-MN reorganization after staggered double hemisection. a** Schematic shows unilateral injection of BDA (red circle) into the T9 (right) spinal cord in mice with T7 (right) and T12 (left) lateral hemisections. **b** Representative image shows an anti-GFAP stained horizontal section of the spinal cord after staggered hemisections. Arrowheads indicate the midline. Scale bar = 600 μm. **c** Representative images show BDA+ dPST axonal distribution in transverse sections of the lumbar spinal cord (L3–L4) in different experimental groups. Box plots indicate BDA+ dPST fiber density index (left side relative to the right side of the spinal cord) and mean density of dPST fibers per area of left L3–L4. Scale bar = 500 μm. **d** High magnification images show the BDA+ dPST axons innervating CTB-labeled (CTB+) lumbar MNs (left L3–L4). Arrowheads indicate the appositions between dPST axons and MNs in a 3D view. Box plots indicate the mean density of dPST fiber per area and the number of colocalized puncta in the left ventral horn of L3–L4. Scale bar = 20 μm. **e** Schematic shows the tcMMEP response recorded from left gastrocnemius muscle, were evoked by a magnetic field applied to the motor cortex. Representative tcMMEP signals are plotted among three groups at 6 weeks after the staggered lesions. The plots report the average latency and peak-to-peak amplitude of tcMMEPs among three groups. **f** Schematic and representative stacked traces show the trajectory of left hindlimb endpoints between groups at 6 weeks after staggered lesions. Box plots indicate average step lengths and heights between groups. Data are presented as box plots with center lines indicating medians, boxes representing 25th to 75th percentiles, and whiskers representing data points within 1.5 times the IQR; $n = 5$–6 biologically independent animals per group; *$P < 0.05$; **$P < 0.01$; ***$P < 0.001$. One-way ANOVA followed by Tukey's post-hoc test. tcMMEPs transcranial magnetic motor-evoked potentials, CTB cholera toxin subunit B. Source data are provided as a Source Data file.

## Discussion

In this study, we found that, upon recovery of lumbar MNs, mediated by the NT-3 therapy, the sprouting and sparing of descending pathways projecting to the lumbar spinal cord was essential for locomotor recovery after a T9 contusion. We then found that the contusive SCI interrupted both the CST and RST projections at the site of injury, but spared the descending pro-priospinal system which conveyed the corticospinal and rubrospinal signals down to the lumbar MNs. Pathway-selective

silencing of neural transmission in three, spared, descending pathways (i.e. the serotoninergic, dopaminergic, and dPST) revealed that the dPST was the primary pathway that functionally accounts for the NT-3-mediated functional recovery. Lastly, our time-course study revealed that NT-3-stimulated propriospino-MN neurocircuit reorganization occurred mainly via promoting regrowth of atrophied dendrites rather than preventing existing dendrites from injury-induced dendritic degeneration. To our knowledge, this is the first study to dissect functional pathways

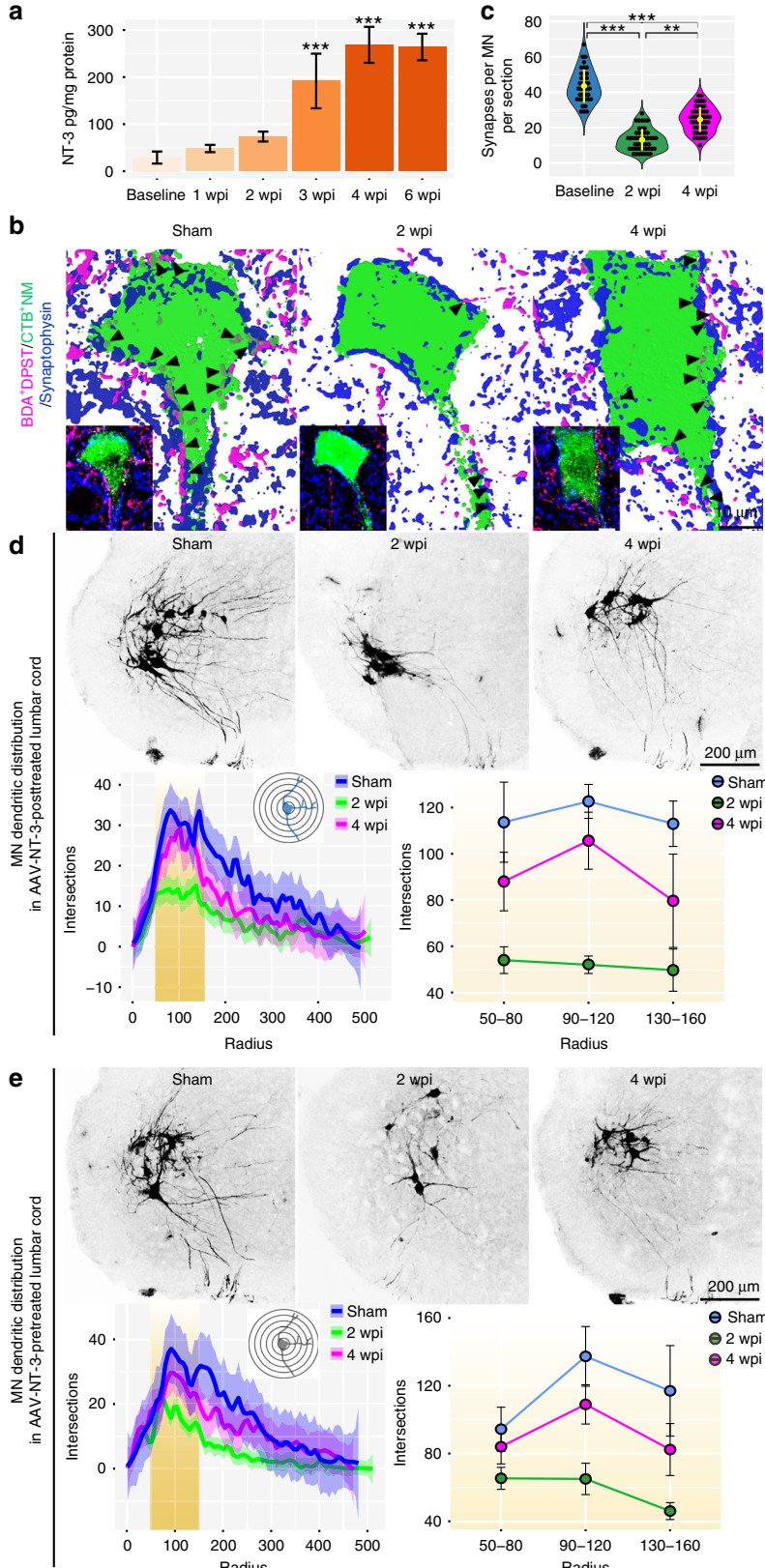

that innervate the lumbar neurocircuit, as well as the first study to reveal a critical role of descending propriospinal-lumbar MN circuit for enabling motor control after a rostral, contusive SCI.

Locomotor movements in vertebrates are not only generated by activities in spinal neurons themselves but also regulated by descending pathways from supraspinal areas[1,2]. SCIs can

profoundly interrupt supraspinal projections to the lumbar circuit leading to locomotor deficits. Although lesioned descending axons have limited ability to reconnect to their targets by long-distance regeneration[33], the reorganization of spared corticospinal[14], reticulospinal[34], and intraspinal[15,35] projections can provide an extensive capacity for functional recovery which

**Fig. 6 NT-3 induced neuroplasticity of lumbar MNs following contusion. a** Bar graph shows local NT-3 expression levels, measured by ELISA, in lumbar cord before and after contusions followed by AAV-NT-3 treatment. Data are presented as mean ± SEM; $n = 3–4$ mice per each time point; ***$P < 0.001$ vs baseline (before SCI). One-way ANOVA followed by Tukey's post-hoc test. **b** Representative images show synapse-like contacts (arrowheads) between BDA$^+$ dPST axons and CTB$^+$ lumbar MNs in sham and contusive mice with AAV-NT-3 post-treatment at 2 and 4 wpi. Scale bar = 10 μm. **c** Quantitative analysis of the number of triple-labeled appositions between BDA$^+$ dPST, CTB$^+$ MNs, and synaptophysin per each MN per section among three groups. Dots in violin plots represent triple appositions from 3–4 mice in each group. The yellow lines indicate mean ± SD. **$P < 0.01$, ***$P < 0.001$; one-way ANOVA followed by Tukey's post-hoc test. **d** Representative images show lumbar MN dendritic complexity in the three experimental groups. Curve plot represents sholl analysis of MN morphology among groups. Thick lines represent the average distributions of MN dendrites for each experimental group ($n = 3$ mice per group); shaded areas represent 95% confidence intervals. Line plot details the difference in the dendritic complexity of MNs at distance from somas between 50 and 160 μm. Data are presented as mean ± SEM. Two-way ANOVA followed by Tukey's post-hoc test. **e** Representative images show MN dendritic complexity in sham and T9 contusion mice with AAV-NT-3 pre-treatment and MN dendritic morphology examined at 2 and 4 wpi. Curve plot represents sholl analysis of MN dendritic distribution in the three groups. Thick lines represent the average distributions of MN dendrites among groups ($n = 3$ mice per group); shaded areas represent 95% confidence intervals. Line plot details the difference in the dendritic complexity of lumbar MNs at distances from somas between 50 and 160 μm. Data are presented as mean ± SEM. Two-way ANOVA followed by Tukey's post-hoc test. Source data are provided as a Source Data file.

represents an attractive target for therapeutic manipulation. In line with this, a crucial question in this study is whether descending pathways are responsible for NT-3-induced functional recovery, and if so, which pathway plays a major role. To answer this question, we took advantage of the complete transection model followed by retrograde transport of NT-3 to lumbar MNs. We found that, although the NT-3 treatment reversed MN dendritic atrophy, it failed to produce significant motor recovery in the absence of descending motor control, after the T9 total transection. This suggests that, in addition to preventing lumbar MN atrophy, restoring descending motor control of lumbar MN circuitry is required for NT-3-mediated recovery after SCI. Thus, enhancing the plasticity of descending pathways (particularly that of the contusion-spared, NT-3-sensitive dPST) and restoring caudal-to-injury lumbar MN dendritic morphology and synaptic formation are indispensable for NT-3-induced locomotor recovery after a rostral thoracic SCI.

Our in vitro and in vivo NT-3 ELISA results showed that NT-3 was released from lumbar MNs to the local environment. Examination of the axonal distribution of the CST, RST, and dPST showed that NT-3 had no spatial diffusion effect on axon sprouting or regeneration at or above the lesion site. The effect of NT-3 is confined within the lumbar spinal cord, indicating that only residual pathways that reach the lumbar region account for NT-3-induce recovery. By contrast, our moderate contusive SCI profoundly disrupted both the CST and RST projections to lumbar MNs. Despite no residuals, their descending neuro-transmissions were preserved downstream[14,35], evidenced by electrophysiological recordings, suggesting alternative routes that conduct corticospinal and rubrospinal comments down to the lumbar MN circuit.

Both our previous[7] and current data showed that mono-aminergic (dopaminergic and serotonergic) pathways were spared after a spinal cord contusion and that they responded to the NT-3 treatment by increasing local collateral sprouting. However, silencing synaptic transmission of these neurons by a dual virus technique failed to alter locomotor performance. Since a small number of descending monoaminergic neurons were targeted in the silencing experiment, two possibilities exist concerning the functional roles of the spared monoaminergic pathways. One possibility is that the NT-3 treatment induced monoaminergic axons to increase their terminal sprouting in the lumbar cord post-injury, but they are not functionally engaged in locomotor recovery. Alternatively, the residual monoaminergic pathways may contributed to NT-3-induced recovery, but the double-infection technique applied here infected too few monoaminergic neurons to show an functional effect. For the second possibility, this limitation was not observed in sham animals where many

GFP-positive neurons were found within dopaminergic and serotonergic brain regions. In that group, we still did not observe significant locomotor deficits during Dox administration, suggesting that these pathways may play a minimal role in the recovery after SCI. In addtion to monoaminergic pathways, literature reported that the reticulospinal tract and the vestibulospinal tract may also be spared after a spinal contusion, and that both pathways show growth response to neurotrophic factor modulation, such as GDNF[36,37] and BDNF[24,38], relay the cortical commands downstream and improve recovery after SCI[14,34]. In our contusion SCI model, as cortical projections may reroute to subcortical regions, and monoaminergic pathways from the brainstem play limited roles, an influence of either reticulospinal tract or vestibulospinal tract, or both, on NT-3-mediated functional recovery could not be excluded. This question could be addressed in future studies using similar approaches as presented in the current study.

Since supraspinal pathways are not directly responsible for NT-3-mediated recovery after SCI, we determined whether this enhancement of recovery was due to an intermediate pathway through dPNs, which has been implicated in the control of forelimb and hindlimb behavior[39–42]. Courtine et al.[15], and Bareyre et al.[35], previously reported that propriospinal-detour-pathways, bypassing supraspinal commands around the lesion site, were able to mediate spontaneous functional recovery and supraspinal control of stepping after interruption of long descending supraspinal pathways in mice. Such an anatomical feature and astonishing plasticity of dPNs have made them attractive targets for treatment after SCI[17,43]. Presently, the accepted medical practices, including activity-based rehabilitation and epidural electrical stimulation, are based upon the concept of activating and promoting extensive reorganization of residual neural pathways that improve locomotion[44–46]. In line with these findings, we found that dPNs that bypass the lesion received their inputs from various descending pathways such as the CST, RST, and cervical dPST, down to the lumbar MNs. These results also corroborate other observations in intact[22,40] and injured spinal cords[34,35]. We then found that residual dPST axonal terminals increased their spouting at the lumbar level and enhanced their synaptic connections with lumbar MNs in response to the retrogradely transported NT-3 in lumbar MNs. Blockade of synaptic transmission through the dPNs perturbed the NT-3-mediated recovery, suggesting that dPNs and the propriospino-lumbar MN circuit contribute to the enhancement of locomotor recovery in mice treated with NT-3 after SCI, and this was confirmed by our staggered double hemisection experiments. Another study notes that such synaptic plasticity and circuit reorganization of PNs after SCI are well conserved across mammals[47]. Experiments in

nonhuman primates and humans show that dPNs contribute to recovery after CST lesions or SCI[48–51]. Therefore, our findings support the idea that NT-3-mediated enhanced functional recovery occurs through the reorganization of the spared dPST. Also, when targeting the plasticity of the dPST, we may ask whether cervical or thoracic dPNs are essential for functional recovery. In our study, we conclude that the thoracic dPNs are essential at least beneficial. However, it would still be interesting to systematically dissect which dPNs play a more significant role, and what will be the consequence if both dPNs are silenced at the same time via dual viral technique in the future.

Neurotrophins are a family of proteins that is essential for the development and maturation of the nervous system, including the regulation of neuronal survival, neurite outgrowth, synaptic plasticity, and neurotransmission. Among them, NT-3 mRNA is highly expressed in the developing spinal cord in MNs but decrease in the adult spinal cord[52,53]. NT-3 is essential for MN survival, target finding, innervation, and synapse formation, mainly during development and early postnatal maturation[52,54]. We hypothesize that vector-induced NT-3 expression in adult MNs may allow us to mimic some of the observations seen during development. In addition, published evidence shows that the expression of TrkC receptor that NT-3 binds with the highest affinity was not limited to MNs, but was seen on the vast majority of neurons throughout the gray matter of spinal cord[52,55,56], suggesting that other populations of neurons also respond to the NT-3 expression. A body of studies also demonstrate that local, sustained expression of NT-3 supports the plasticity of CST[20,57], spinal neuronal survival, and regeneration[58,59]. More importantly, NT-3 can also serve as chemoattractive guidance for regenerated axons to establish their projection, find their targets, and reform appropriate connections[58,60]. Together, the rationales discussed above guided us to take advantage of AAV-NT-3 as a beneficial therapeutic treatment for targeting lumbar MNs after SCI. In the current study, we demonstrate that NT-3 expression via AAV-directed gene therapy restored lumbar MN dendritic morphology, increased remodeling of local neurocircuit, and promoted significant locomotor restoration following a rostral-level thoracic contusion.

Several studies showed that neurotrophins, released from grafts/neurons or provided exogenously, are neuroprotective or pro-regenerative for recovery after SCI[20,21]. Determining the specific role of NT-3 in mediating MN recovery is very important for translating NT-3 treatment into preclinical and/or clinical settings. We addressed this question by assessing the dendritic features of lumbar MNs in response to NT-3 with either pre-treatment (i.e. before contusion) or post-treatment (i.e. after contusion). We observed lumbar MN dendritic withdrawal at 2 wpi and regrowth at 4 wpi after SCI, irrespective of if NT-3 gene therapy was applied as pre- or post-treatment. This result indicates that retrograde transport of NT-3 to the lumbar MNs does not prevent SCI-induced lumbar MN dendritic atrophy, but rather facilitate lumbar MN dendritic regrowth after their initial withdrawal. This mechanistic insight implies that NT-3 gene therapy is potentially applicable to a broad range of patients who are suffering from subacute or long-term chronic SCIs, even after rehabilitation is initiated. As physical activity is capable of stimulating NT-3 synthesis which modulates sensory and motor function[8,61], we speculate that, when combined with physical rehabilitation, AAV-NT-3 gene therapy can further boost neuroplasticity of residual neural pathways and facilitate robust functional recovery.

CNS-directed AAV gene therapies have produced promising pre-clinical and clinical results by the mechanism of inserting, replacing, or deleting genes. Substantial evidence from clinical trials has supported the safety, tolerability, and efficacy of AAV$_2$

as a promising treatment approach for targeting the CNS[62,63]. In addition, NT3 was shown to be safe and well-tolerated in phase I and II clinical trials for CNS disorders[64,65]. However, no clinical studies have been initiated to investigate the effectiveness of NT-3 for promoting motor functional recovery after SCI. In our animal studies[7], we took advantage of the human NT-3 transgene which could effectively be translated into clinical therapy. Thus, our preclinical study paves the way for AAV-NT-3 gene therapy as a potential treatment for the functional restoration of SCI patients.

In conclusion, our study demonstrates that NT-3-stimulated propriospino-lumbar MN neurocircuit improved functional recovery after a rostral-level thoracic contusive SCI. Thus, this study opens new perspectives to guide spinal cord repair interventions in the future.

## Methods

**Animals.** Adult female C57BL/6J mice were purchased from the Jackson Laboratory (8–10 weeks old, 18–22 g body weight). All animal experiments were performed in compliance with all relevant ethical regulations for animal testing and research and in accordance with animal protocols approved by the Institutional Animal Care and Use Committee of Indiana University School of Medicine (IACUC #11011 and #18081) and Institutional Biosafety Committee (IBC #1556). The experimental protocols strictly followed the guidelines of National Institutes of Health (NIH) on the use of laboratory animals. To achieve various goals in this study, we have induced three different thoracic-level SCI models, including complete T9 transection, moderate T9 contusion, and staggered (T7 and T12) lateral hemisection. In each injury model, mice were randomly divided into three experimental groups. Group 1: sham (laminectomy) + saline; Group 2: SCI + AAV-GFP; Group 3: SCI + AAV-NT-3. Either saline or AAV-GFP or AAV-NT-3 (1 µl) was bilaterally injected into pre-demyelinated sciatic nerves in each group animal after surgery, respectively[7].

**AAV viral preparation and injection.** Recombinant adeno-associated virus 2 (AAV$_2$) carrying GFP (AAV-GFP, $1.0 \times 10^{13}$ viral particles/ml; Temple University, Philadelphia, USA) was used as a control. Human NT-3 subcloned into an AAV vector cassette under the control of the chicken-beta actin (CBA) promoter and containing a polyadenylation signal from human B-globin gene was prepared by the Dr. George Smith Lab (AAV-NT-3; $1.0 \times 10^{12}$ viral particles/ml; Temple University, Philadelphia, USA)[7]. Five days after transient demyelination with lysolecithin (Fischer Scientific, Pittsburg, PA; 0.5 ml in 1% phosphate-buffered saline (PBS)), both sides of the sciatic nerves were injected with 1 µl of either AAV-GFP or AAV-NT-3 using a 10-µl Hamilton syringe attached to a finely pulled capillary and the injection tip was held in place for an additional 2 min to avoid the viral leakage.

**Surgical procedures and animal care.** For all surgical procedures, animals were anesthetized with ketamine (17.2 mg/ml)/xylazine (0.475 mg/ml)/acepromazine (0.238 mg/ml). After surgery, mice received subcutaneous administration of buprenorphine (0.01–0.05 mg/kg every 8–12 h for 3 days) for pain release and 0.5 ml of saline for hydration. The mice were then returned to clean home cages that were partially placed on a heating pad until they fully recovered from the anesthesia. Manual bladder expression was performed twice daily until the bladder emptying by reflex was established.

To produce a spinal cord transection model at T9, a laminectomy was exposed at the 9th thoracic vertebral level, the dura was ruptured with a 30-gauge needle, and the cord was transected with Noyes spring scissors (Fine science tools, Canada) throughout the entire width and depth of the spinal cord. In all cases, a customized microknife was used to retrace the lesion to ensure the completeness of the lesion.

To produce a contusive SCI model at T9, mice were placed on their ventral surface in a U-shaped stabilizer[66]. After a T9 laminectomy, mice received a T9 contusion using the Louisville Injury System Apparatus (LISA, Louisville, KY) with a 0.5-mm diameter tip at a velocity of 1.0 m/s. The severity and consistency of the injury were verified by checking the bruise on the spinal cord, the injury parameters provided by the LISA software[66], and initial behavioral assessments. For bilateral pyramidotomy, the mouse was placed in a supine position and a midline incision was made to the ventral neck to expose the trachea. Blunt dissection was then operated until the ventral surface of the skull was reached. To gain access to the underlying pyramid, a bilateral craniotomy was made in basioccipital bone by using a micro drill, which was attached to steel burr with a ball-size of 0.6 mm (Stoelting, Item No. 58610). After the dura was incised longitudinally by a 32-gauge needle, a bilateral cut was then made in the pyramid. The pyramidotomy on each side was ~1.0 mm wide spanning the width of the pyramid and 0.5 mm deep with a modified fine scalpel perpendicular to the basilar artery to interrupt the descending CST axons.

To bilaterally transect the RST projections, a laminectomy was made to expose the dorsolateral aspect of the spinal cord as well as the dorsal roots of C3 and C4 (ref. [25]). The dura matter was opened by a bent 32-gauge needle and the lateral part

of the dorsal lateral funiculus was then transected in a depth of 1 mm from the spinal cord dorsal surface with a modified fine scalpel mounted on a blade holder.

The procedure of T7 and T12 staggered double hemisection was similar to that described elsewhere[15]. Briefly, a midline incision was made over the thoracic vertebrae, followed by a T7–12 laminectomy. For the lateral hemisection, we inserted a custom-made half-open needle (32-gauge) from the midline down into the spinal cord to guide both a micro-scissors and a scalpel to interrupt the half-side of the spinal cord through the midline.

**Anatomical tracing**. To anterogradely label the projection of CST axons to the spinal cord, mice with craniotomy over the motor cortex were placed in a ste-reotaxic frame (Stoelting, USA) and injected with 0.5 µl of 10% BDA w/v in sterile saline (MW 10,000, Invitrogen, USA) into one of 10 total sites (5 sites/side) with the following coordinates: mediolateral (ML) coordinate: 1.5 mm lateral to the bregma; anteroposterior (AP) coordination from the bregma: −1.0, −0.5, 0, −0.75, and 1.5 mm; dorsoventral (DV) coordination: 0.5 mm from the cortical surface. The BDA was delivered by a 10-µl Hamilton syringe, at a rate of 0.1 µl/min, controlled by a digital stereotactic injector (Item: 51709, Stoelting Co. USA). After each injection was completed, the injector tip was left in the cortex for an addi-tional 5 min to ensure that the BDA adequately penetrated the tissue without leaking. Two weeks later, mice were anesthetized and perfused with 4% paraformaldehyde for further anatomical analysis.

For anterograde tracing of RST[25], a 10% BDA solution was injected bilaterally into the red nucleus (3.15 and 3.65 mm posterior to the bregma, 0.6 mm lateral to the midline, all at a depth of 3.6 mm from the cortex surface). Each injection site was administered 0.5 µl of BDA within 5 min and the glass needle was left in place for an additional 5 min before removal. Animals were sacrificed 2 weeks later.

For bilaterally tracing cervical and thoracic dPST axons, we stereotaxically injected 10% BDA solution into the intermediate gray matter of C5 and T7 cord segments. The injection coordinates were as follows: 0.5 mm lateral to the midline and 0.9 mm deep from the dorsal surface of the C5 cord segment; 0.3 mm lateral to the midline and 0.6 mm deep from the dorsal surface of the T7 cord segment, respectively. For unilaterally labeling thoracic dPST axons in the staggered double hemisected spinal cord (T7 and T12), 2 × 0.3 µm of 10% BDA was injected into the right side of T9/T10 segments with the same coordinates as described in the labeling of T7 thoracic dPNs. The second BDA injection was placed 0.5 mm caudal to the first one. Mice were kept for an additional 5–7 days before sacrifice.

To retrogradely label lumbar MNs, either AAV-GFP or Alexa Fluor 488 conjugates of CTB (2% solution, Invitrogen, USA) was injected bilaterally into both of the sciatic nerves of mice at 6 weeks following SCI. Mice were kept for 7 days after receiving the tracer injection.

For transsynaptic tracing of MN-related dPNs, PRV-152 (gift from Dr. Lynn W. Enquist's lab at Princeton University) was bilaterally administered at three different sites of the gastrocnemius muscle, using a 10-µl Hamilton syringe fitted with a 30-gauge needle, according to the existed protocol[25]. At 24, 48, and 72 h post-inoculation, animals were perfused with 0.9% saline followed by 4% paraformaldehyde, to determine the different order of PRV-labeled neurons in the spinal cord.

**Stimulus-triggered EMG recordings**. To measure evoked EMG potentials in tibialis anterior muscles, a craniotomy was made to allow the electrodes to access the M1 motor cortex or the red nucleus. The penetration points in the motor cortex were chosen from 1 to −1 mm caudal, 0.5 mm lateral to the bregma with a depth of 0.6 mm from the surface, while the penetration points in the red nucleus were chosen from −3 to −4 mm caudal, 0.6 mm lateral to the bregma with a depth of 3.6 mm from the surface. Electrical stimulation was delivered using a tungsten bipolar elec-trode (Microprobes, USA) via an isolated stimulator (A-M system, USA). The elec-trical stimulation is consisted of 7 bipolar pulses, each being 200 µs in duration, at a frequency of 333 Hz, and repeated every 2 s. The stainless-steel wire was inserted into the contralateral tibialis muscle for recording the EMG signal. The EMG activities were amplified by a differential AC amplifier (model 1700, A-M system, USA) and converted as an analog signal via an analog-to-digital converter. For each single penetration site, 20 stimulation sweeps were recorded. The non-responsive stimulus site which failed to produce any response within a range of current from 0.5 µA to 500 µA was switched to next at a distance of 0.3 mm along with the rostrocaudal axis relative to bregma. Signals with movement artifacts were excluded. Only animals with stable EMG responses were included in this experiment.

**Double virus-mediated silencing experiments**. HiRet-TRE-EGFP.eTeNT (2.19 × 10[7] TU/ml) and AAV₂-CMV-rtTAV16 (AAV-Tet/On, 1.2 × 10[13] GC/ml) were constructed and pre-tested in Dr. George Smith's lab[23]. To assess the lumbar cord, musculature was cleared from the T11–T13 vertebral bodies, and laminec-tomies were performed to expose the L1–L4 spinal cord. HiRet-TRE-EGFP.eTeNT was injected into the ventral horn of L2–L4 segments with three separate sites on each side of the spinal cord. Injections were made at 0.6 mm lateral to the midline and a depth of 1.1 mm from the dorsal surface, with a volume of 0.5 µl per each site. For targeting the cell bodies of dPNs, serotoninergic neurons or dopaminergic neurons, AAV-Tet/On was injected into the T5–T7 spinal cord, raphe nuclei, or the midbrain A11 region, respectively. Injection coordinates for T5–T7 spinal cord

were 0.3 mm lateral to the midline and 0.6 mm deep from the dorsal surface of the spinal cord (six injections total, 0.5 µl per injection). Injection coordinates for raphe nucleus were −5.4, −5.9, −6.4 mm caudal to bregma, 0 mm along the midline, and −5.6 mm ventral from the surface of the cerebellum (three injections total, 0.5 µl per injection). Injection coordinates for the midbrain A11 region were −1.9 mm caudal, ±0.6 mm mediolateral to the bregma, and −4.6 mm deep from the dorsal surface of the midbrain (two injections total, 0.5 µl per injection). All injections were done using a digital mouse stereotaxic instrument (Stoelting, USA) fitted with a beveled, pulled glass needle, at a speed of 0.1 µl/min, the needle was held in place for 5 min before being slowly retracted. After tracer injections, all animals were maintained for an additional 3 weeks before the administration of Dox was initi-ated. The Dox (2 mg/ml) was added into the drinking water with 1% sucrose. Mice from all three experimental groups (sham, SCI + AAV-GFP, and SCI + AAV-NT-3) were tested by using BMS[7,67], grid walking[68], and rotarod[7,68] behavioral assessments before Dox administration (baseline), during Dox administration (4 days after initiation), and 7 days after the cessation of Dox administration.

**Behavioral assessments**. The BMS test was performed to evaluate the overall basic locomotor performance[7,67]. Briefly, mice were placed in an open field (dia-meter: 42 inches) for 4 min per each trial. Two trained observers who were blinded to experimental groups judged the BMS score on a scale of 0–9 (0, complete hind limb paralysis; 9, normal locomotion), which is based on hind limb movements made in an open field including hind limb joint movement, weight support, plantar stepping, coordination, paw position, and trunk and tail control.

For the grid walking test, a mesh metal grid setup (12 mm × 12 mm) was used to monitor skilled locomotor performance[68]. Each mouse was placed in the center of the mesh grid and the hindlimb locomotion on a grid was evaluated by two experimenters who were blinded for the animal groups over 3 min, with one person counting the total number of footsteps while the other counting the number of times that each of the hindlimbs fell through the grid holes.

For the rotarod test (IITC Life Science, USA)[7], the pre-trained animals were placed in the accelerating lanes of rotarod from 3 to 15 revolutions per minute (rpm) over 2 min with a total of five trials per session. The latency of mice maintaining themselves on the rotating rod was recorded based on when they fell off for each trial and averaged into a final score per session.

**NT-3 ELISA**. The NT-3 ELISA assay was performed to assess NT-3 expression levels before and after SCI[7]. Briefly, the lumbar spinal cord was quickly dissected from the mouse and frozen in a dry ice/ethanol bath and kept at −80 °C for future analysis. NT-3 ELISA (Human NT-3 DuoSet ELISA, R&D Systems, USA) was performed with duplicates according to the manufacturer's protocol to measure the secreted NT-3 protein in vivo. NT-3 concentration was calculated based on a linearized standard curve ranging from 0 pg/ml to 600 pg/ml.

**BDA staining**. The 30-µm-thick sections were incubated with avidin-biotin-peroxidase (Vectastain R.T.U. Elite ABC Reagent, SAT700, PerkinElmer, USA) for 1 h. Biotinyl tyramide (PerkinElmer) was diluted to 1:100 with amplification diluent, then placed on the sections for an additional hour after the wash. The sections were then incubated with Extra-Avidin@ TRITC (Sigma, USA, 1:200 in 0.3% PBST) for 2 h. Lastly, the sections were washed again and coverslipped with SouthernBiotech Fluoromount-G mounting medium (Cat. No. 0100-01, SouthernBiotech).

**Immunofluorescence staining**. For immunofluorescence staining, 30-µm-thick frozen sections were incubated with the following primary antibodies: rabbit anti-serotonin (5-HT, 1:2000, Sigma #S5545), rabbit anti-TH (1:500, Millipore #AB152), rabbit anti-tropomyosin receptor kinase C (TrkC, 1:500, Abcam ##ab43078), and mouse anti-synaptophysin (Syn, 1:1000, Millipore #MAB5258), rabbit anti-glia fibrillary acidic protein (GFAP, 1:1000, Abcam, #ab7260) and goat anti-cholera toxin B (CTB, 1:2000, List Biological Laboratories, #703). Signal was detected with the corresponding second antibodies conjugated to Alexa Fluor 405 or 488 or 594 fluorescence (1:1000, Invitrogen).

**Sholl analysis**. To measure the detailed topologies of the lumbar MN dendrites among different experimental groups across different time points, the sholl analysis was applied on GFP-labeled MNs by using an open-source software, ImageJ (NIH, USA). Briefly, a series of circles with increasing radii (10 µm apart) centered in the MN pools. The number of the intersections of MN dendrites with each concentric circle at an incremental distance was counted automatically by ImageJ, and the associated curve plot demonstrated the number of intersections at various distances from the MN cell body.

**The topography of CST and RST terminations**. To analyze the CST and RST topographic distribution within the gray matter in the lumbar enlargement, BDA-positive images were grouped in Photoshop software (Adobe System, USA) where the size of the gray matter, and associated contours, were normalized to standar-dized dorsoventral and mediolateral lengths of the spinal cord. Subsequently, the images were aligned and processed with the Threshold and Skeletonization func-tions in ImageJ, and then converted to skeleton images. The digitized x–y

coordinates of extremes of the branches in the results represent the axon terminal distributions and densities in a given area of the section. All branch information was then transferred into MATLAB (MathWorks, Natick, MA) for further processing to generate the regional density heatmaps for each section by custom codes, with the pure red representing the highest axon density and the pure blue representing the lowest axon density for each section.

**Data quantification**. To measure the dPST axon density, we acquired BDA-labeled images in the field of the ventral horn of the spinal cord using a 20× objective (Fig. 3d) or 60× oil objective (Fig. 5d). The serial images were imported into ImageJ and the mean fiber integrated density was calculated. We adjusted the density values by dividing the area of the selected subregion of the images.

To count close appositions between BDA-labeled dPST axons and CTB-labeled lumbar MNs, the Z-stack images from cross-sections were acquired using a 60× oil objective of a confocal fluoview microscope (Olympus, Japan) in the lumbar spinal cord (L2–L4). We examined all apposed puncta in a 3D-view manner. Only appositions in all three dimensions were considered as valid candidates to be counted (indicated as arrowhead in Supplementary Fig. 4e), with the intention of eliminating other appositions just in one or two dimensions (indicated as arrow in Supplementary Fig. 4e). We counted all valid appositions of BDA-labeled dPST axons with 2–4 CTB-labeled MNs within an unbiased virtual courting space in each section. In each animal, we counted 4–6 sections. The data were expressed as either appositions per MN per section (Supplementary Fig. 4f) or appositions per MN per animal (Fig. 5d).

We determine the propriospino-MN synaptic connections by quantification of the number of colocalized puncta stained for BDA-labeled dPST terminals, CTB-labeled MN, and a presynaptic marker, synaptophysin. The synapse numbers were determined by the number of triple appositions in the counting frame by defined tissue space. As described above, we unbiasedly counted 2–4 MNs in each section and 4–6 sections per animal. The data were finally expressed as synapses per MN per section (Fig. 6c).

For analysis of TrkC expression density, images of the ventral horn region were captured with a 40× objective from transverse sections stained with TrkC immunohistochemistry. The mean integrated density was measured in Image J with the same methods described above. To analyze the dPST axon terminals co-expressing TrkC, we counted the number of double-labeled puncta in TrkC immunoreactive and BDA-labeled sections in Photoshop. The absolute number of co-localized puncta in each group was then normalized to dPST fiber density to allow comparisons of NT-3-induced effect on TrkC expression between groups.

To count the double-infected dPNs, the T4–T8 region of the spinal cord was dissected out and the serial cryosections were prepared and collected. We counted and summated the number of GFP-positive cells, when present, in all sections using Photoshop.

To count the double-infected cells around the B1, B2, and B3 regions of the caudal brainstem, sagittal and coronal brainstem sections were prepared and all GFP-positive cells were counted in the raphe nuclei and other neighboring reticular formation regions. To estimate the number of double-infected serotoninergic neurons in the brainstem, we counted the number of GFP-positive cells co-localized with 5-HT-immunoreactive within all regions of interest.

To count the double-infected cells within the A11 region, the total number of GFP-positive cells in midbrain sections were quantified from bregma −1.5 mm to −2.5 mm. To assess the double-infected dopaminergic neurons in the A11 region, we counted the number of GFP-positive cells co-localized with TH-immunoreactive neurons in the A11 region of all sections.

The EMG responses we collected for statistical quantification in each experimental group met the following three criteria: (1) the EMG response contained a peak occurring at the normal latency for the target muscle; (2) such a peak had to be present in response to at least 4 of the 10 stimuli given; and (3) the EMG peaks had to be larger than the other peaks from the background. The peak-to-peak amplitudes and onset latencies of the contralateral EMG responses were measured in Clampfit (Molecular Devices, USA) and further plotted in R studio (Open source software).

To assess the density of CST, RST, and monoaminergic fibers in the lumbar cord, serial immunoreactive sections were traced and reconstructed automatically with the Imaris software using the FilamentTracer function (BitPlane AG, Switzerland). The total filament length and the number of branches in the reconstructions were analyzed to estimate the axon densities in each experimental group.

To measure the step length and height of the left hindlimb in sham mice and the mice that received staggered hemisections. The mice were recorded when walking through a narrow custom-built plexiglass trough (5 cm wide by 40 cm long) via a high-speed camera. We then used Kinovea (Open source software, https://www.kinovea.org) to manually trace the motion of all straightforward steps of left rear paws. Any twist or backward or non-stepping steps were excluded from the final quantification. Step length was measured by the distance between the center of subsequent left rear paws; step height was defined by the maximum step height of left rear paw lifting from the ground.

**Statistical analysis**. Statistical parameters including the definitions and exact values of the number of animals, deviations, P values, and the types of the statistical

tests were reported in the figures and corresponding figure legends. Statistical analysis was carried out using GraphPad Prism software (version 7.00, La Jolla, California, USA) or R studio. Comparisons between the experimental groups were performed by one-way ANOVA or two-way ANOVA with Tukey's post-hoc test. $P < 0.05$ was considered statistically significant.

**Reporting summary**. Further information on research design is available in the Nature Research Reporting Summary linked to this article.

## Data availability
All relevant data as well as computer code supporting the findings of the current study are available from the corresponding authors upon reasonable request. The source data underlying Figs. 1c, d, 2e, f, 3d, f, 4b–d, 5c–f, 6a, c–e, and Supplementary Figs. 2b, d, 4d, f, 5d, 7b, 8b, d, 9b, d are provided as a Source Data file.

## Code availability
All relevant computer codes supporting the findings of the current study are available from the corresponding authors upon reasonable request.

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

## Acknowledgements

This work was supported in part by NIH 1R01 100531, 1R01 NS103481, Merit Review Award I01 BX002356, I01 BX003705, I01 RX002687 from the U.S. Department of Veterans Affairs, Indiana State Department of Health (ISDH, Grant# 19919), Mari Hulman George Endowment Funds (X-M.X), DOD grant and Shriners (GMS). Additional thanks to Baylen Ravenscraft for his editing suggestions.

## Author contributions

Q.H. designed and performed all the experiments, and analyzed the data; J.D.O. performed behavior assessments, analyzed the data, and revised the manuscript; N.-K.L. revised the manuscript; Z.R. captured the images and analyzed the data; W.W., Y.P.Z., and C.B.S. provided technical assistance for the SCI models; Y.X., W.Q., Y.W. and H.D. performed behavioral assessments; G.M.S. provided the virus, instructed the dual-virus silencing experiment and revised the manuscript; X-M.X. is the corresponding author who conceived, designed, and directed the project; Q.H. and X-M.X. wrote the manuscript.

## Competing interests

The authors declare no competing interests.
