## [Peer Review File · Nature Communications]

Reviewers' Comments:

Reviewer #1:

Remarks to the Author:

This paper uses multiple approaches to identify neurons and pathways responsible for forming relay connections from above to below incomplete spinal cord injuries (SCI). The authors previously reported that NT3 delivered via AAV vectors to peripheral nerves attracts spared axons to contact motor neurons and thereby increases connections and improves various functional outcome measures. In the present study the authors find that the major contributors to spared neural connections that continue to project past the injury after their contusion SCI are propriospinal neurons that are located within the spinal cord a few segments above the injury or in the cervical region. In contrast, descending connections from various brain regions (cortex, brainstem) did not contribute significantly to spared connections. As far as I can tell, the work appears to have been rigorously conducted and is of high quality. The experiments are well designed and properly controlled and have sufficient replicates to have statistical power. The data presented look good and are convincing. The findings are important on several levels and should be of interest to a broad audience interested in restoring function after spinal cord injury and more generally in understanding the mechanisms that can promote the regeneration of functional neural circuits. I have no concerns regarding technical aspects of the study. It seems very thorough. I have only a few suggestions for edits regarding the text.

Specific comments:

1) About half-way through the results, around page 7 lines 176 onward, the experimental model switches from evaluations long projecting propriospinal neurons in the cervical area injury for anterograde tracing studies to evaluating short projecting propriospinal neurons in the thoracic area around T6 just above by retrograde tracing studies after contusion injuries which it seems were always at T9 for both anterograde and retrograde tracing. Studying both sets of neurons can be valid in their own right, but the switch is never explained. Why use one set of neurons for anterograde another for retrograde analyses? Some explanation should be provided and perhaps a bit of discussion on differences between long cervical and short thoracic propriospinal neurons.

2) NT3 is a pleiotropic growth factor with many targets and many functions and effects. The authors should provide a bit of background and discussion with literature citations on why they think that providing NT3 as a retrograde factor to motor neurons would induce motor neurons to attract more connections by propriospinal neurons. Why did they try this strategy, and what background evidence (literature) is there about how it might work? Is this a mechanism that is documented to function during development?

Reviewer #2:

Remarks to the Author:

The article by Han et al is interested in dissecting the effects of NT-3 and follows on a recent report demonstrating that retrograde delivery of NT-3 to motoneurons attenuate SCI-induced lumbar motoneurons dendritic atrophy and improved functional recovery. In this report the authors go further in their investigations to demonstrate that (i) residual projections to lumbar motoneurons are necessary to produce leg movements, (ii) that spared propriospinal connections are more important than supraspinal motor tracts to mediate NT-3 induced recovery and (iii) that NT-3 treatment promotes motoneuron dendritic regrowth. The paper is well written, very well illustrated and overall provides new insights on NT-3-induced spinal remodeling of axonal connections. The material and method section relative to the quantifications should be more detailed to allow a correct interpretation of the data.

I have the following additional remarks that need to be addressed:

Figure 3 panel b: no scale bar is provided

Figure 3 panel c: this panel is supposed to show close appositions between residual propriospinal fibers and motoneurons. What the images show is a higher density of propriospinal terminals after Nt-3 treatment but no appositions. While it could be expected that there is more appositions the images do not allow appreciating this point as the lack of quantification. Investigating appositions would require higher magnification images and quantifications of these appositions in single planes and 3D views. This should be presented in the paper and detailed in the material and methods as this is an important and interesting point.

Figure 4 panel a: What is the rationale to now investigate thoracic propriospinal neurons instead of the cervical ones investigated before? It is not clear why the experiment has not followed logically with using the cervical propriospinal neurons. Please comment.

Figure 5 a1: what is the hole at T9? There is no contusion here.

Figure 5 a3: here again there is no obvious colocalisation of puncta as could be defined in the image. Higher magnification images would be needed to carry on such analysis as the use of single plans and 3D images. The detail of such analysis would also need to be added to the material and methods section.

Figure 6 c: was this analysis corrected for motoneuron size? Couldn't find the information on the material and method. This is quite important to interpret the images in b which do not provide the information themselves.

Reviewer #3:

Remarks to the Author:

It has been difficult to obtain direct evidence of specific axonal pathways involved in a return of function after spinal cord injury, with most previous reports ending up being correlational at best. The present study provides considerable tract tracing, pathway silencing, and electrophysiological data that propriospinal neurons rostral to the injury have a significant role in improved locomotor function after a lower thoracic contusion injury of adult mice, to the exclusion of corticospinal, rubrospinal serotonergic and dopaminergic axons. Part of the enhanced recovery is attributed to NT-3 mediated modulation of lumbar motoneuron circuitry and propriospinal neuron sprouting, and yet while there is observed regrowth of the motoneuron dendritic field there is no evidence that this has a role in return of function.

There are three major conclusions but the second one about identifying the important role of the propriospinal-motoneuron circuitry for recovery of motor control is the strongest and most convincingly argued by the data. The first conclusion, that spared axons are required for recovery of motor function (i.e. axons do not recover in a complete transection model) is not surprising. The third conclusion is that motoneuron dendrites atrophy after SCI but regrow if motoneurons are transfected with AAV-NT3. This is interesting but does not seem to fit in with the observation of propriospinal-motoneuron circuitry and locomotor function after SCI. At least there is no data provided to link dendrite restructuring to a functional role. It is not clear that dendritic regrowth is indispensable for locomotor recovery (line 391).

Overall the experimental design is outstanding, with precise tracing of pathways, clever use of a dual neuron silencing approach and incorporation of electrophysiological stimulation to determine whether cortical or red nucleus neuron stimulation was involved in direct stimulation of lumbar motoneurons after SCI. This is a strong body of work will be of use to the SCI regeneration-plasticity field.

A point of concern is the absence of discussion of reticulospinal and vestibulospinal pathways and their role in rodent locomotion. It is not clear that the impact of these pathways can be discounted without being tested. It is stated in Abstract and Discussion that the spared descending propriospinal pathway and not other pathways account for recovery. Also it is stated (line 289) that the propriospinal-MN circuit reorganization is functionally required for recovery, but this study does not indicate that it is sufficient for recovery. Other pathways may be involved. It would

appear that both ReST and VST pathways are minimally damaged by the contusion injury (Suppl Figure 3) but there may be some response to local increase in NT-3 as seen with other pathways. This should be discussed at least.

Minor issues:

Line 494 - change pre-myelinated to pre-demyelinated

Line 618 - change trail to trial

John Houle

Point-by-point Responses to the Reviewers Comments

Reviewer #1 (Remarks to the Author)

Overall comments:

As far as I can tell, the work appears to be have been rigorously conducted and is of high quality. The experiments are well designed and properly controlled and have sufficient replicates to have statistical power. The data presented look good and are convincing. The findings are important on several levels and should be of interest to a broad audience interested in restoring function after spinal cord injury and more generally in understanding the mechanisms that can promote the regeneration of functional neural circuits. I have no concerns regarding technical aspects of the study. It seems very thorough. I have only a few suggestions for edits regarding the text.

Response: We thank the reviewer for his/her strong enthusiasm for this manuscript.

Specific comments:

1) About half-way through the results, around page 7 lines 176 onward, the experimental model switches from evaluations long projecting propriospinal neurons in the cervical area injury for anterograde tracing studies to evaluating short projecting propriospinal neurons in the thoracic area around T6 just above by retrograde tracing studies after contusion injuries which it seems were always at T9 for both anterograde and retrograde tracing. Studying both sets of neurons can be valid in their own right, but the switch is never explained. Why use one set of neurons for anterograde another for retrograde analyses? Some explanation should be provided and perhaps a bit of discussion on differences between long cervical and short thoracic propriospinal neurons.

Response: The reviewer raised an important question which was also shared by reviewer #2. In an early study, we found that retrograde transport of AAV-NT-3 to lumbar motoneurons (MNs) stimulated the sprouting of spared cervical dPST terminals in lumbar MN pools ¹. The increased dPST fibers were in close apposition with CTB-labeled MNs, indicating that NT-3 treatment reinforced residual propriospino-MN circuit reorganization. To confirm the NT-3-modulated propriospino-MN connections, we took advantage of a time-dependent, multi-transsynaptic virus, PRV, to retrogradely label the MN-related descending propriospinal neurons (dPNs). At 72 h, we found that, despite the apparent disruption of propriospino-MN connections, there was still a significant increase in PRV-labelled dPNs at the T7 level in SCI mice after the AAV-NT-3 treatment, as compared to SCI control mice with AAV-GFP treatment (**Supplementary Fig. 5c, d**). Notably, these dPNs above the lesion relayed descending commands, such as CST, RST, and cervical dPST, down to the lumbar motor circuit (**Supplementary Fig. 6**). In contrast to dozens of PRV transsynaptically-labeled neurons at the thoracic level, few, if any, labeled dPNs were found in the cervical level in contusive mice with either AAV-GFP or AAV-NT-3 treatment at 72 h post-inoculation when the time is sufficient for viral propagation to label plenty of neurons in cervical segment in uninjured sham mice (**Response Fig.1, inserted below, also revised**

Supplementary Fig. 5c). This result suggests that more direct synaptic connections between the thoracic dPN-MN circuit might exist as compared to the cervical dPN-MN circuit. With consideration of the feasibility and efficacy of doxycycline-induced dual-viral system, the finding of more PRV-labeled, MN-related dPNs in the thoracic level than the cervical level, is the primary reason that prompted us to use thoracic dPNs for functionally dissecting the role of descending propriospino-MN circuit in NT-3-mediated recovery. Secondly, although the T9 contusion can result in large-scale alterations of spared dPST that relay the supraspinal commands downstream, the considerable reorganization may occur around or near the lesion site. Therefore, in comparison to the cervical dPST, the thoracic spared dPST axons may contribute more directly to the recovery of function after a thoracic contusion. In the dPN selective silencing experiment, we choose to inject AAV/Tet-On into T5-T7, but not T8, due to the concern of its diffusion across the lesion site.

Although thoracic dPNs above the lesion site are considered to be more functionally relevant to the contribution of NT-3-mediated motor recovery, we can not exclude the engagement of the cervical dPNs. The approach of PRV-labeled dPNs is in a time-dependent manner. The contusive animals may require more time to label MN-related dPNs in the cervical level than sham mice due to the lesion that may lead to severe disruptions of the direct dPST-MN circuit. Therefore, future work determining whether cervical propriospinal neurons also play a role in NT-3-mediated locomotor recovery would be interesting; if they do, it would also be interesting to know which dPNs play a more significant role, and what will be the consequence if both dPNs are silenced at the same time.

We have added data analysis to **Supplementary Fig. 5c** and incorporated such discussion on **page 10 and 18**.

2) NT3 is a pleiotropic growth factor with many targets and many functions and effects. The authors should provide a bit of background and discussion with literature citations on why they think that providing NT3 as a retrograde factor to motor neurons would induce motor neurons to attract more connections by propriospinal neurons. Why did they try this strategy, and what background evidence (literature) is there about how it might work? Is this a mechanism that is documented to function during development?

Response: This is also an excellent comment made by the reviewer. We agree that introducing more background and discussion of NT-3 will provide a more detailed rationale behind why we chose NT-3 as a retrogradely transported trophic factor to reverse MN atrophy and restore local motor circuits after a rostral SCI.

The lumbar MNs are the final common pathway for the motor output of the hindlimbs. They can be impaired by a direct injury to the lumbar cord or by an indirect injury that occurred at levels above the lumbar cord. When an SCI occurs at the above lumbar levels (namely above-level injury), the lumbar MNs are not directly injured by the trauma, but they undergo profound

dendritic atrophy and synaptic stripping from denervated supraspinal and propriospinal axons. Such altered lumbar MN morphological and synaptic changes could result in impaired motor outputs to hindlimb muscles and therefore impaired locomotor functions. Whereas most SCI studies have been focused on the regeneration or protection of injured spinal cord at the site of injury, few studies have explored how modulation of lumbar MN circuitry would affect pathological and functional consequences after an above-level SCI. Our previous study showed that NT-3 could be a promising restorative treatment to recover MN atrophy and locomotor deficits after an above-level SCI. In the present study, we aim to investigate the anatomical and functional mechanism of NT-3-enhanced locomotor recovery.

Neurotrophins are a family of proteins that regulate neuronal survival, neurite outgrowth, synaptic plasticity, and neurotransmission. Among them, NT-3 mRNA is highly expressed in the developing spinal cord in motor neurons but decreased in the adult spinal cord^{2,3}. NT-3 is essential for MN survival, target finding, innervation, and synapse formation, mainly during development and early postnatal maturation^{2,4}. Therefore, we hypothesize that vector-induced NT-3 expression in MNs may allow us to mimic some of the observations seen during development, which can either protect MNs from atrophy or promote dendritic regrowth of these MNs after SCI.

It should be noted that the expression of TrkC receptor that NT-3 binds with the highest affinity was not limited to motoneurons, but was seen on the vast majority of neurons throughout the grey matter of spinal cord^{2,5,6}, suggesting that various other populations of neurons also respond to the NT-3 expression. A body of studies has demonstrated that local, sustained expression of NT-3 supports the plasticity of corticospinal tract⁷⁻⁹, spinal neuronal survival, and regeneration^{10,11}. More importantly, NT-3 can also serve as chemotropic guidance for regenerated axons in establishing their projection, selecting their targets and reforming appropriate connections^{10,12,13}. Together, the rationales we mentioned above channel the hypothesis that retrogradely-transported NT-3 to lumbar MNs may be a beneficial strategy that is able to attenuate SCI-induced lumbar MN dendritic atrophy, attract more connections from spared descending pathways, and, therefore, promote locomotor functional recovery after a thoracic contusion. We have added a new discussion with new literature citations on why NT-3 was chosen as a treatment for SCI (page 18-19).

Reviewer #2 (Remarks to the Author)

Overall comments:

The article by Han et al is interested in dissecting the effects of NT-3 and follows on a recent report demonstrating that retrograde delivery of NT-3 to motoneurons attenuate SCI-induced lumbar motoneurons dendritic atrophy and improved functional recovery. In this report the authors go further in their investigations to demonstrate that (i) residual projections to lumbar motoneurons are necessary to produce leg movements, (ii) that spared propriospinal connections are more important than supraspinal motor tracts to mediate NT-3 induced recovery and (iii) that NT-3 treatment promotes motoneuron dendritic regrowth. The paper is well written, very well illustrated and overall provides new insights on NT-3-induced spinal remodeling of axonal

connections. The material and method section relative to the quantifications should be more detailed to allow a correct interpretation of the data.

Response: We thank the reviewer #2 for his/her positive comment on our manuscript. We have provided more detailed information on the material and method section relative to the quantifications to allow a correct interpretation of the data.

Specific comments:

(1) Figure 3 panel b: no scale bar is provided.

Response: Scale bar has been added to the panel b in Figure 3.

(2) Figure 3 panel c: this panel is supposed to show close appositions between residual propriospinal fibers and motoneurons. What the images show is a higher density of propriospinal terminals after Nt-3 treatment but no appositions. While it could be expected that there is more appositions the images do not allow appreciating this point as the lack of quantification. Investigating appositions would require higher magnification images and quantifications of these appositions in single planes and 3D views. This should be presented in the paper and detailed in the material and methods as this is an important and interesting point.

Response: We appreciate the reviewer's comment. We found that NT-3 treatment stimulated a higher density of propriospinal terminals in lumbar MN pools, but we did not directly quantify the close appositions between propriospinal terminals and lumbar MNs. We agree with the reviewer's suggestion to quantify these appositions in single planes and 3D views via higher magnification images. Sections of the lumbar spinal cords (L2-L4) with both CTB-labeled MNs and BDA-labeled dPST axons were imaged every fourth 40- μ m cross-sections. All the apposition data described were acquired in the ventral horns (MN pools) of these sections using a 60x oil objective confocal fluoview microscope (Olympus, Japan). To determine the appositions, we studied the co-labeling of MNs with propriospinal axons in a 3D view. The criteria for quantification are that only the apposition observed in all three dimensions is considered a valid candidate to be counted (indicated as **arrowhead** in **Supplementary Fig 4e**), with the intention of eliminating other appositions just in one or two dimensions (indicated as **arrow** in **Supplementary Fig 4e**). We counted all valid appositions of BDA-labeled dPST axon terminals with 2-4 CTB-labeled MNs within an unbiased virtual courting space in each section. In each animal, we counted 4-6 sections. We have incorporated this quantification in **Supplementary Fig 4 e and 4f**. With this data, we confirmed that retrograde transport of AAV-NT-3 to lumbar MNs increased the terminal BDA-labeled dPST fiber density in lumbar MN pools, as compared to AAV-GFP treatment. Importantly, the increased dPST terminals in the ventral horn intermingled and apposed with CTB-labeled MNs (**Supplementary Fig 4e, f**). This result suggests that retrograde transport of AAV-NT-3 to lumbar MNs stimulates the reorganization of spared dPST connections with lumbar MN pools following contusive SCI. As the reviewer suggested, we have presented and highlighted the detailed methods in the materials and methods section (**page 28-29**).

(3) Figure 4 panel a: What is the rationale to now investigate thoracic propriospinal neurons instead of the cervical ones investigated before? It is not clear why the experiment has not followed logically with using the cervical propriospinal neurons. Please comment.

Response: The reviewer raised an important question which was also raised by reviewer #1. Please refer to our response to comment #1 of reviewer #1 (page 10 and 18).

(4) Figure 5 a1: what is the hole at T9? There is no contusion here.

Response: Figure 5 a1 is a representative image of an anti-GFAP stained horizontal spinal section showing the lesion borders of staggered hemisections (T7 and T12) as well as two BDA injection sites (T9/T10) for unilaterally labeling the thoracic dPST on the right side of the spinal cord. Therefore, the lesion area at T9, which is surrounded by a strong expression of reactive astrocytes, is the center of BDA injection sites. While we realize that BDA injections appear to induce a strong GFAP response which spreads over the T9/T10 segments. Two possibilities may cause such an occurrence. First, the puncture of the injector tip, as well as respiration and other movements from the animal during tracer injections can cause the deformation of spinal tissue, leading to additional contusion beyond the injection site. Second, for the thoracic dPST tracing, we sacrificed the mice 5-7 days post-injection. Such a time period is an acute stage when excessive reactive astrocytes are generated in response to CNS damages¹⁴. Together, these reasons probably explain why we observed that substantial GFAP response spread over the injection site.

(5) Figure 5 a3: here again there is no obvious colocalisation of puncta as could be defined in the image. Higher magnification images would be needed to carry on such analysis as the use of single plans and 3D images. The detail of such analysis would also need to be added to the material and methods section.

Response: This is a similar issue raised in comment #2 of reviewer #2 (see above). We agree with the reviewer that it would be more evident to identify the colocalized puncta in the images with a 3D view. Therefore, we replaced the representative images with images in a 3D view. All images used for quantification of appositions between BDA-labeled dPST fibers and CTB-labeled neurons are Z-stacks of confocal images acquired using a 60X oil-immersive objective in sequential mode to avoid crosstalk between channels. We also follow the same criteria as described before in the execution of apposition analysis. To be consistent with previous presentations, the data were expressed as appositions per MN per animal. As the reviewer suggested, we also added a detailed analysis in the materials and methods section (page 28-29).

(6) Figure 6 c: was this analysis corrected for motoneuron size? Couldn't find the information on the material and method. This is quite important to interpret the images in b which do not provide the information themselves.

Response: Thanks for this comment! The analysis in Figure 6c was not corrected for motoneuron size. We determine the propriospino-MN synaptic connections by quantification of the number of colocalized puncta stained for both BDA-labeled dPST axonal terminals, CTB-labeled MNs

and a presynaptic marker, synaptophysin. The synaptic numbers were determined by counting the number of triple appositions in the counting frame by defined tissue space. As we described in the previous response, we unbiasedly counted 2-4 MNs in each section and we counted 4-6 sections per each animal. The synapse numbers were finally expressed as the number of synapses per MN per section. We added a new detailed description of synapse quantification in the materials and methods section (page 29).

Reviewer #3 (Remarks to the Author)

Overall comments:

It has been difficult to obtain direct evidence of specific axonal pathways involved in a return of function after spinal cord injury, with most previous reports ending up being correlational at best. The present study provides considerable tract tracing, pathway silencing, and electrophysiological data that propriospinal neurons rostral to the injury have a significant role in improved locomotor function after a lower thoracic contusion injury of adult mice, to the exclusion of corticospinal, rubrospinal serotonergic and dopaminergic axons. Part of the enhanced recovery is attributed to NT-3 mediated modulation of lumbar motoneuron circuitry and propriospinal neuron sprouting, and yet while there is observed regrowth of the motoneuron dendritic field there is no evidence that this has a role in return of function.

There are three major conclusions but the second one about identifying the important role of the propriospinal-motoneuron circuitry for recovery of motor control is the strongest and most convincingly argued by the data. The first conclusion, that spared axons are required for recovery of motor function (i.e. axons do not recover in a complete transection model) is not surprising. The third conclusion is that motoneuron dendrites atrophy after SCI but regrow if motoneurons are transfected with AAV-NT3. This is interesting but does not seem to fit in with the observation of propriospinal-motoneuron circuitry and locomotor function after SCI. At least there is no data provided to link dendrite restructuring to a functional role. It is not clear that dendritic regrowth is indispensable for locomotor recovery (line 391).

Overall the experimental design is outstanding, with precise tracing of pathways, clever use of a dual neuron silencing approach and incorporation of electrophysiological stimulation to determine whether cortical or red nucleus neuron stimulation was involved in direct stimulation of lumbar motoneurons after SCI. This is a strong body of work will be of use to the SCI regeneration-plasticity field.

Response: We thank Dr. Houle for his insightful comments on this manuscript.

A point of concern is the absence of discussion of reticulospinal and vestibulospinal pathways and their role in rodent locomotion. It is not clear that the impact of these pathways can be discounted without being tested. It is stated in Abstract and Discussion that the spared descending propriospinal pathway and not other pathways account for recovery. Also it is stated (line 289) that the propriospinal-MN circuit reorganization is functionally required for recovery, but this study does not indicate that it is sufficient for recovery. Other pathways may be involved. It would appear that both ReST and VST pathways are minimally damaged by the contusion

injury (Suppl Figure 3) but there may be some response to local increase in NT-3 as seen with other pathways. This should be discussed at least.

Response: We thank Dr. Houle for raising this excellent question! As we have not investigated ReST and VST yet in the present study, we agree with Dr. Houle's comment that we cannot exclude the anatomical and functional contributions of these two important pathways in response to NT-3-induced recovery. Therefore, we added a paragraph, described below, to discuss such possibilities in the revised manuscript:

“In contusive animals, it appears that both ReST and VST are also partially damaged and survive the lesions. Published evidence suggests that those two pathways show growth response to neurotrophic factor modulation, such as GDNF^{15,16} and BDNF^{17,18}, relay the cortical commands downstream and improve the recovery after SCI¹⁹⁻²¹. As cortical projections reroute to the brainstem, monoaminergic pathways play limited roles, and the influence of both ReST and VST on NT-3-mediated functional recovery could not be excluded in the current study. This question could be addressed in the future by assessing the axonal plasticity of ReST and VST in the lumbar cord with or without NT-3 treatment and by individually dissecting the functional roles of these pathways in hindlimb motor recovery via the dual-viral system as used in this study.” (added to the discussion session on **page 17**).

Minor issues:

Line 494 - change pre-myelinated to pre-demyelinated

Line 618 - change trail to trial

Response: The minor issues have been corrected in the revision.

Related References:

- 1 Wang, Y. *et al.* Remodeling of lumbar motor circuitry remote to a thoracic spinal cord injury promotes locomotor recovery. *eLife* **7**, doi:10.7554/eLife.39016 (2018).
- 2 Keefe, K. M., Sheikh, I. S. & Smith, G. M. Targeting Neurotrophins to Specific Populations of Neurons: NGF, BDNF, and NT-3 and Their Relevance for Treatment of Spinal Cord Injury. *Int J Mol Sci* **18**, doi:10.3390/ijms18030548 (2017).
- 3 Ernfors, P. & Persson, H. Developmentally Regulated Expression of HDNF/NT-3 mRNA in Rat Spinal Cord Motoneurons and Expression of BDNF mRNA in Embryonic Dorsal Root Ganglion. *European Journal of Neuroscience* **3**, 953-961, doi:10.1111/j.1460-9568.1991.tb00031.x (1991).
- 4 Morcuende, S., Muñoz-Hernández, R., Benítez-Temiño, B., Pastor, A. M. & de la Cruz, R. R. Neuroprotective effects of NGF, BDNF, NT-3 and GDNF on axotomized extraocular motoneurons in neonatal rats. *Neuroscience* **250**, 31-48, doi:<https://doi.org/10.1016/j.neuroscience.2013.06.050> (2013).
- 5 Bradbury, E. J., King, V. R., Simmons, L. J., Priestley, J. V. & McMahon, S. B. NT-3, but not BDNF, prevents atrophy and death of axotomized spinal cord projection neurons.

- European Journal of Neuroscience* **10**, 3058-3068, doi:10.1046/j.1460-9568.1998.00307.x (1998).
- 6 Johnson, R. A., Okragly, A. J., Haak-Frendscho, M. & Mitchell, G. S. Cervical Dorsal Rhizotomy Increases Brain-Derived Neurotrophic Factor and Neurotrophin-3 Expression in the Ventral Spinal Cord. *The Journal of Neuroscience* **20**, RC77-RC77, doi:10.1523/JNEUROSCI.20-10-j0005.2000 (2000).
- 7 Ruitenber, M. J. *et al.* NT-3 expression from engineered olfactory ensheathing glia promotes spinal sparing and regeneration. *Brain* **128**, 839-853, doi:10.1093/brain/awh424 (2005).
- 8 Schnell, L., Schneider, R., Kolbeck, R., Barde, Y. A. & Schwab, M. E. Neurotrophin-3 enhances sprouting of corticospinal tract during development and after adult spinal cord lesion. *Nature* **367**, 170-173, doi:10.1038/367170a0 (1994).
- 9 Weishaupt, N. *et al.* Vector-induced NT-3 expression in rats promotes collateral growth of injured corticospinal tract axons far rostral to a spinal cord injury. *Neuroscience* **272**, 65-75, doi:10.1016/j.neuroscience.2014.04.041 (2014).
- 10 Wan, G., Gomez-Casati, M. E., Gigliello, A. R., Liberman, M. C. & Corfas, G. Neurotrophin-3 regulates ribbon synapse density in the cochlea and induces synapse regeneration after acoustic trauma. *Elife* **3**, doi:10.7554/eLife.03564 (2014).
- 11 Liu, Y. *et al.* NT-3 promotes proprioceptive axon regeneration when combined with activation of the mTor intrinsic growth pathway but not with reduction of myelin extrinsic inhibitors. *Exp Neurol* **283**, 73-84, doi:10.1016/j.expneurol.2016.05.021 (2016).
- 12 Alto, L. T. *et al.* Chemotropic guidance facilitates axonal regeneration and synapse formation after spinal cord injury. *Nat Neurosci* **12**, 1106-1113, doi:10.1038/nn.2365 (2009).
- 13 Hunanyan, A. S., Petrosyan, H. A., Alessi, V. & Arvanian, V. L. Combination of chondroitinase ABC and AAV-NT3 promotes neural plasticity at descending spinal pathways after thoracic contusion in rats. *J Neurophysiol* **110**, 1782-1792, doi:10.1152/jn.00427.2013 (2013).
- 14 Liddel, S. A. & Barres, B. A. Reactive Astrocytes: Production, Function, and Therapeutic Potential. *Immunity* **46**, 957-967, doi:10.1016/j.immuni.2017.06.006 (2017).
- 15 Dolbeare, D. & Houle, J. D. Restriction of axonal retraction and promotion of axonal regeneration by chronically injured neurons after intraspinal treatment with glial cell line-derived neurotrophic factor (GDNF). *J Neurotrauma* **20**, 1251-1261, doi:10.1089/089771503770802916 (2003).
- 16 Weishaupt, N., Li, S., Di Pardo, A., Sipione, S. & Fouad, K. Synergistic effects of BDNF and rehabilitative training on recovery after cervical spinal cord injury. *Behavioural Brain Research* **239**, 31-42, doi:<https://doi.org/10.1016/j.bbr.2012.10.047> (2013).
- 17 Jin, Y., Fischer, I., Tessler, A. & Houle, J. D. Transplants of fibroblasts genetically modified to express BDNF promote axonal regeneration from supraspinal neurons following chronic spinal cord injury. *Exp Neurol* **177**, 265-275, doi:10.1006/exnr.2002.7980 (2002).
- 18 Xu, X. M., Guénard, V., Kleitman, N., Aebischer, P. & Bunge, M. B. A Combination of BDNF and NT-3 Promotes Supraspinal Axonal Regeneration into Schwann Cell Grafts in Adult Rat Thoracic Spinal Cord. *Experimental Neurology* **134**, 261-272, doi:<https://doi.org/10.1006/exnr.1995.1056> (1995).

- 19 Asboth, L. *et al.* Cortico-reticulo-spinal circuit reorganization enables functional recovery after severe spinal cord contusion. *Nat Neurosci*, doi:10.1038/s41593-018-0093-5 (2018).
- 20 Bachmann, L. C. *et al.* Deep brain stimulation of the midbrain locomotor region improves paretic hindlimb function after spinal cord injury in rats. *Science translational medicine* **5**, 208ra146, doi:10.1126/scitranslmed.3005972 (2013).
- 21 Filli, L. *et al.* Bridging the gap: a reticulo-propriospinal detour bypassing an incomplete spinal cord injury. *J Neurosci* **34**, 13399-13410, doi:10.1523/jneurosci.0701-14.2014 (2014).

Reviewers' Comments:

Reviewer #1:

Remarks to the Author:

In this revised manuscript, the authors edited the paper well and have appropriately taken care of my comments and concerns. They have also responded to requests by other reviewers. I have no further comments or concerns and I continue to find the paper interesting and convincing.

Reviewer #2:

Remarks to the Author:

The paper is now strengthened and suitable for publication.

Reviewer #3:

Remarks to the Author:

Clearly the importance of propriospinal-motoneuron circuitry for recovery is presented in this study but it is important that the question raised about possible involvement of descending pathways in locomotor recovery not examined in this study was added in the Discussion.

Something that was raised but not asked for directly was clarification about the role of dendritic sprouting related to locomotor recovery. There appears to be a correlation but not direct evidence that sprouting has a role or is essential for locomotor recovery. It is not clear that the answer to the question asked on lines 88-90 is both. I don't see the data that NT-3 mediates locomotor recovery by promoting dendritic regrowth (lines 103-105 and 391-394). Is dendritic sprouting indispensable?

Minor: A word may be missing from line 419 and line 451

REVIEWERS' COMMENTS:

Reviewer #1 (Remarks to the Author):

In this revised manuscript, the authors edited the paper well and have appropriately taken care of my comments and concerns. They have also responded to requests by other reviewers. I have no further comments or concerns and I continue to find the paper interesting and convincing.

Response: We truly appreciate this reviewer's constructive comments to improve the quality of our work.

Reviewer #2 (Remarks to the Author):

The paper is now strengthened and suitable for publication.

Response: We also appreciate the reviewer's insightful comments, particularly on the difference between cervical and thoracical propriospinal neurons.

Reviewer #3 (Remarks to the Author):

Clearly the importance of propriospinal-motoneuron circuitry for recovery is presented in this study but it is important that the question raised about possible involvement of descending pathways in locomotor recovery not examined in this study was added in the Discussion.

Something that was raised but not asked for directly was clarification about the role of dendritic sprouting related to locomotor recovery. There appears to be a correlation but not direct evidence that sprouting has a role or is essential for locomotor recovery. It is not clear that the answer to the question asked on lines 88-90 is both. I don't see the data that NT-3 mediates locomotor recovery by promoting dendritic regrowth (lines 103-105 and 391-394). Is dendritic sprouting indispensable?

Response: We appreciate Dr. Houle's further insightful comments particularly on the role of MN dendritic sprouting in locomotor recovery. We agree that addressing this issue is important to clarify major principles and mechanisms underlying NT-3-mediated locomotor recovery.

Our moderate T9 contusion model showed that lumbar MNs were not directly affected by a rostral level injury, but they underwent a profound dendritic withdrawal and synaptic stripping due to supraspinal denervation. In this SCI model, although contusions often spared some descending pathways, including the descending propriospinal tract (dPST), the contusive mice

only exhibited limited ability to perform consistent locomotion, suggesting that, without recovery of lumbar MNs, it is difficult to restore locomotion with some spared descending pathways. Our T9 transection model eliminated all descending projections to the lumbar spinal cord. In this condition, retrograde transport of AAV-NT-3 into the lumbar MNs reversed their dendritic atrophy by sprouting but failed to improve locomotor recovery, as compared to the AAV-GFP group. This suggests that, in the absence of spared descending innervation, the lumbar MN dendritic sprouting alone is insufficient to restore locomotion. Together, these results indicate that therapeutic strategies aimed at modulating the plasticity of both lumbar MN and spared descending pathways may enable better functional recovery. Indeed, our results showed that retrogradely transported NT-3 to lumbar MNs enhanced propriospino-MN circuit reorganization, as reflected with an increase in sprouting of both MN dendrites and dPST terminals as well as synaptic formations of neural circuits, which leads to functional improvement after SCI.

In the context of NT-3 gene therapy via the sciatic nerve-related route following SCI, we believe MN dendritic sprouting plays an indispensable role in propriospino-MN circuit reorganization which functionally accounts for NT-3-mediated locomotor recovery. In the absence of descending innervation (eg, in the T9 transection model), MN dendritic sprouting alone is insufficient and plays a limited role in locomotor improvement.

In lines 103-104, we previously concluded that NT-3 mediates locomotor recovery via promoting dendritic regrowth rather than by preventing dendritic atrophy. This conclusion, as Dr. Houle pointed out, is less accurate because we did not compare the behavioral changes between NT-3 pre-treatment and NT-3 post-treatment. We have now used “NT-3 mediates MN recovery” instead of “NT-3 mediates locomotor recovery” in our conclusion (lines 101-102).

In lines 391-394, we summarized that NT-3 released from lumbar MNs worked as a local modulatory factor on lumbar motor circuit remodeling, and that it did not expand over to the lesion site with an effect on axon sprouting or regeneration of damaged descending pathways.

In summary, our results demonstrate that NT-3-induced MN dendritic regrowth/sprouting may not be a correlation but is indispensable for propriospino-MN circuit reorganization which is required for NT-3-mediated functional recovery.

Minor: A word may be missing from line 419 and line 451

Response: Thanks for your kind reminder! The missing words have been added in the revised manuscript. Please see lines 425-426 (previous line 419) on page 18 as well as lines 458-459 (previous line 451) on page 19 highlighted by yellow.